# Spatial entropy drives the maintenance and dissemination of transferable plasmids

Wenzhi Xue[1,3], Juken Hong [iD][1,3], Runmeng Zhao[2], Huaxiong Yao[1], Yi Zhang[1], Zhuojun Dai[1] & Teng Wang [iD][1✉]

## Abstract

The dissemination of transferable plasmids, a major type of mobile genetic elements (MGEs), is one main driver of antibiotic resistance outbreaks. While the plasmid persistence condition in well-mixed environments has been extensively studied, most microbiota in nature are spatially heterogeneous. However, our knowledge regarding how spatial landscape shapes plasmid maintenance and dissemination remains limited. Here we establish a theoretical framework describing plasmid spread over a metacommunity of multiple patches. By analyzing the gene flow dynamics on randomly generated landscapes, we show that plasmid survival and dispersal are dictated by a simple feature of the landscape, spatial entropy. Reducing entropy speeds up plasmid range expansion and allows the global maintenance of many plasmids that are predicted to be lost by classic theories. The entropy's effects are experimentally validated in *E. coli* metacommunities transferring a conjugative plasmid. We further examine a vast collection of prokaryotic genomes and show that prokaryotes from low-entropy environments indeed carry more abundant MGEs and antibiotic resistance genes. Our work provides critical insights into the management and control of antimicrobial resistance.

**Keywords** Horizontal Gene Transfer; Plasmid; Spatial Entropy; Antibiotic Resistance; Biofilm
**Subject Categories** Computational Biology; Microbiology, Virology & Host Pathogen Interaction

## Introduction

Plasmids, a major type of mobile genetic elements (MGEs), are important components of microbial metagenomes. Plasmids encode diverse biological functions like metabolic capabilities, pathogenic virulence, antimicrobial resistance, or traits to cope with environmental stresses (Frost et al, 2005). Understanding the quantitative principles governing plasmid persistence and

dissemination in complex environments is critical for our capability to predict, control and engineer microbiome functions (Lawson et al, 2019; Wang and You, 2020). The persistence conditions of plasmids have been extensively studied in well-mixed microbial communities (Bergstrom et al, 2000; Brockhurst and Harrison, 2022; Lopatkin et al, 2017; MacLean and San Millan, 2015; Stewart and Levin, 1977), largely driven by the growing threats from antibiotic resistance (Castañeda-Barba et al, 2024; Larsson and Flach, 2022). Theoretical analysis predicts that a sufficiently high transfer rate can overcome a plasmid's fitness cost and allow its stable persistence (Lopatkin et al, 2017; Stewart and Levin, 1977). However, the general applicability of these predictions has been debated (Bergstrom et al, 2000; Lili et al, 2007; Lopatkin et al, 2017; Simonsen, 1991; Stalder and Top, 2016). Many empirical estimates in natural populations suggested that the required transfer rates could be too high to be biologically realistic (Bergstrom et al, 2000; Lili et al, 2007; Lopatkin et al, 2017; Stalder and Top, 2016).

In nature, the spatial homogeneity of microbial populations can be readily disturbed. For instance, by forming surface-attached biofilms or free-floating aggregates (Fig. 1A,B), bacterial cells cluster in a self-produced matrix of extracellular polymeric substances, resulting in the increase of local bacterial densities compared with their planktonic counterparts (Flemming et al, 2016; Schlomann and Parthasarathy, 2021). In many environments, nutrients are unevenly distributed (Fig. 1C), which creates microhabitats with varying resource availabilities and leads to the enrichment of bacteria in some areas while deprivation in others (Dal Bello et al, 2021; Reese et al, 2018). In soil, dehydration promotes the transient fragmentation of the aqueous phase among soil particles (Fig. 1D), altering bacterial distribution in the microscale habitats (Tecon et al, 2018). For aquatic microbiota, microplastics from pollution can also bring about spatial heterogeneity by enabling the colonization of microbes on the surfaces (Fig. 1E) (Liu et al, 2021).

Scale matters in microbial ecology (Ladau and Eloe-Fadrosh, 2019; Singer et al, 2007). In spatially homogeneous habitats, the abundance of a plasmid in each local area is representative of its global abundance. In physically structured metacommunities, however, the global maintenance of a plasmid depends not only on its persistence capability in each local population, but also on its efficiency to disseminate across different areas. Such disseminations

[1]Key Laboratory of Quantitative Synthetic Biology, Shenzhen Institute of Synthetic Biology, Shenzhen Institutes of Advanced Technology, Chinese Academy of Sciences, Shenzhen 518055, China. [2]School of Mathematics, Jilin University, Changchun 130012, China. [3]These authors contributed equally: Wenzhi Xue, Juken Hong.
✉E-mail: t.wang1@siat.ac.cn

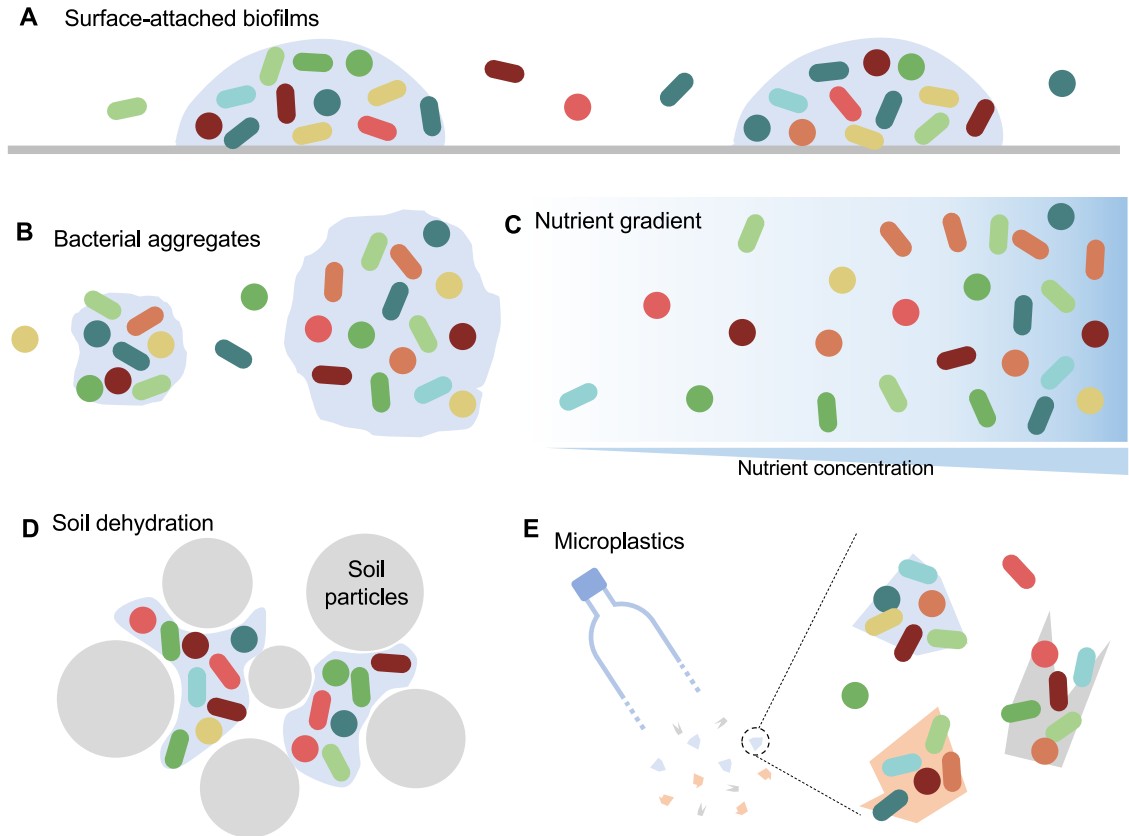

**Figure 1.  Processes that shape the spatial landscapes of microbial communities in natural environments.**

(**A**) By forming biofilms, bacterial cells are densely embedded in a self-produced matrix adherent to a surface. (**B**) Many bacteria are bound to non-attached, free-floating aggregates, which also break the spatial homogeneity of the population. (**C**) Nutrient concentration determines bacterial growth rates. The uneven distribution of nutrients leads to the spatial heterogeneity of cell density along the nutrient gradient. (**D**) Soil dehydration promotes the transient fragmentation of the aqueous phase among soil particles, leading to a heterogeneous distribution of cells in the microscale habitats. (**E**) The spatial landscapes of microbiota in soil or aquatic environments can also be changed by microplastics, the surface of which can be colonized by many bacteria.

are increasingly found in various environments and often associated with antibiotic resistance outbreaks (Berglund, 2015; Jiang et al, 2017; Pinilla-Redondo et al, 2021). The spatial landscape, by shaping the frequency of cell-to-cell contact, will inevitably affect the survival and spread of transferable plasmids. However, our knowledge regarding the general laws underlying this interplay remains limited, due to the intrinsic complexities of the physical structures of microbiota and the lack of quantitative analysis (Castañeda-Barba et al, 2024; Slater et al, 2008). Such knowledge is crucial in many scenarios, especially for the risk evaluation, predictive control and reversal of antibiotic resistance genes (ARGs) at broad scales (Castañeda-Barba et al, 2024; Kim and Cha, 2021; Martínez et al, 2015; Slater et al, 2008; Zhang et al, 2022).

In this work, we established a theoretical framework that described the plasmids' dissemination over a metacommunity composed of multiple patches. By simulating the gene flow dynamics on various randomly generated landscapes, we showed that the maintenance and spread of a plasmid were determined by a simple feature of the landscape, spatial entropy (Wang and Zhao, 2018). In physics, entropy measures the disorder of a system. Here, we used this concept to quantify the spatial heterogeneity of

microbial populations. Reducing entropy promoted the plasmid dissemination and allowed the global maintenance of many plasmids below the 'limits' predicted by classic theories. These effects arose from the negative correlation between entropy and the average plasmid transfer rate. We experimentally validated this relationship in metacommunities of *E. coli* strains transferring a conjugative plasmid. To further validate the predictions, we also examined a vast collection of sequenced prokaryotic genomes and showed that prokaryotes from low-entropy environments indeed carried more abundant plasmids and ARGs. The genomes from low-entropy environments were also equipped with more transposases and toxin/antitoxin-related genes. These results highlight the fundamental role of spatial landscape in shaping microbial functionality and evolution, and provide critical insights for microbiome engineering.

## Results

We began by establishing a mathematical framework that modeled the horizontal gene transfer dynamics in a metacommunity composed of multiple patches. In microbial ecology, a

metacommunity is defined as a collection of local communities distributed across different habitat patches. Each patch represents a discrete, relatively small area within a larger landscape that supports a local population (Leibold et al, 2004). Patches serve as the fundamental spatial units where ecological processes unfold, and they are often characterized by the unique environmental conditions or resources that can influence the composition and structure of the community residing within them. Patches are interconnected through the dispersal routes of species, facilitating the exchange of organisms and genetic materials across the metacommunity. The concept of patches is crucial for studying how environmental heterogeneity and spatial structure influence biodiversity and ecosystem functioning (Logue et al, 2011).

Our framework accounted for the within-patch population dynamics as well as the plasmid dissemination across patches (Fig. 2A; see Methods for details). We assumed that the population in each patch was well-mixed, where the plasmid-carrying cells (donors) and plasmid-free cells (recipients) competed for resources under the constraint of a maximum carrying capacity $N_m$. The plasmid imposed a fitness effect (described by $\lambda$) on host cell growth. Gene flow occurred within the same patch (with a rate $\eta$) or between the adjacent patches (with the rate $\omega$), when donors transferred the plasmid to recipients. The between-patch plasmid transfer, made possible by bacterial dispersal or immigration, provides opportunities for a plasmid originating from one patch to spread over the entire metacommunity (Fig. 2B,C).

The spatial heterogeneity arises when the maximum carrying capacities differ among patches, and the landscape is defined by the distribution of $N_m$ across space (Fig. 2A). When all kinetic parameters are given, this framework allows us to simulate the spatiotemporal dynamics of plasmid dissemination on arbitrary landscapes (Fig. 2B,C). To illustrate the basic concept, we considered the spread of one plasmid in a space colonized by a single species. Without loss of generality, we constructed meta-communities of $101 \times 101$ patches initially occupied by plasmid-free bacteria and seeded the plasmid in the central patch. Then, the temporal dynamics of plasmid abundance in each patch was predicted by numerical simulations, which allowed us to analyze the traveling speed of the plasmid and its abundance accumulation rate in the entire space (Fig. 2D,E). When calculating the traveling speed, we focused on the furthest straight-line distance that the plasmid had spread across from the seeding point (see Methods for details). The traveling speed reflects the plasmid's capability of range expansion, which is particularly relevant when it comes to evaluating the risk of global dissemination of a mobilizable ARG.

Previous studies focusing on well-mixed populations suggested that the capability of a plasmid to stably maintain itself could be predicted by a simple metric termed persistence potential, which was calculated by combining horizontal transfer rate, plasmid fitness burden, plasmid loss rate and dilution rate (Bergstrom et al, 2000; Lopatkin et al, 2017; Stewart and Levin, 1977; Wang and You, 2020). When the spatial landscape of a metacommunity was given, our simulations with randomized parameters suggested that the traveling speed and the accumulation rate of a transferable plasmid were positively correlated with its persistence potential (Fig. 2F; Appendix Fig. S1; see Methods for details). Therefore, mechanisms contributing to persistence potential, for instance, increasing plasmid transfer rate $\eta$ or reducing fitness burden $\lambda$, will promote the persistence and spread of the plasmid. In contrast, the factors reducing persistence potential, including increasing plasmid fitness burden, segregation loss, and dilution, will inhibit the dissemination and accumulation of the plasmid in the metacommunity.

To analyze the role of spatial landscape on plasmid persistence and dissemination, we used Perlin noise to create a variety of landscapes of maximum carrying capacities (Fig. 3A; see Methods for details) (Etherington, 2022). Originally developed to generate natural-looking visual effects for quantities like clouds or marbles in films and games, Perlin noise is a powerful tool to create randomized ecological landscapes or spatial patterns (Etherington, 2022; Perlin, 1985). We characterized each landscape using two quantifiable terms: spatial period and entropy (Fig. 3A; see Methods for details). In a metacommunity of microbes, spatial periodicity is related to the size of and the distance between the recurring structures like biofilms, aggregates or plastispheres. As a key concept in physics and chemistry, entropy measures the disorder of a system. Here, we borrowed this concept to quantify the spatial heterogeneity of microbial populations (see Methods for details). The value of spatial entropy is closely related to the magnitude of bacterial density fluctuations across space. High spatial entropy is typically observed in well-mixed populations, whereas low entropy indicates significant heterogeneity in microbial distribution. In this context, the spatial entropy of a metacommunity is independent of its mean cell density. Entropy measures the evenness of density distribution rather than the average density value across the space (Appendix Fig. S2). Reducing entropy creates not only areas with higher local densities but also areas with lower local densities (Appendix Fig. S2). While it is well established that increased density enhances plasmid transfer by promoting cell-to-cell contact, the impact of changing spatial entropy on plasmid dissemination and persistence remains less understood.

By controlling two independent parameters in the Perlin noise algorithm, we generated diverse landscapes covering wide ranges of spatial periods and entropies (Fig. 3A; see Methods for details). When varying spatial periods and entropies, we ensured that the average value of the maximum carrying capacity across the metacommunity remained constant. This allowed us to disentangle the effects of spatial periodicity and entropy from those of mean density. On each landscape, we seeded a plasmid in the central patch and analyzed its abundance accumulation rate ($u$) in the entire metacommunity as well as its traveling speed ($v$) by numerical simulations (Fig. 3B). Our results suggested that while the $u$ and $v$ values were not significantly affected by spatial periodicity, reducing spatial entropy sped up the accumulation and dissemination of the MGE (Fig. 3C–F). Changing the total number of patches in the metacommunity led to the same prediction (Appendix Fig. S3). This conclusion was not affected by the between-patch plasmid transfer rate, either (Appendix Fig. S4). Simulations on a highly simplified metacommunity composed of only two patches also suggested that reducing entropy promoted the overall abundance of the plasmid (Appendix Fig. S5). Here, we assumed plasmid dissemination mediated by continuous microbial dispersal among different patches. This assumption was not critical for our prediction, either. We considered an array of patches undergoing periodic mixing between different patches (Appendix Fig. S6A). In this setup, local populations in each patch grow independently without cross-patch dispersal until the next mixing

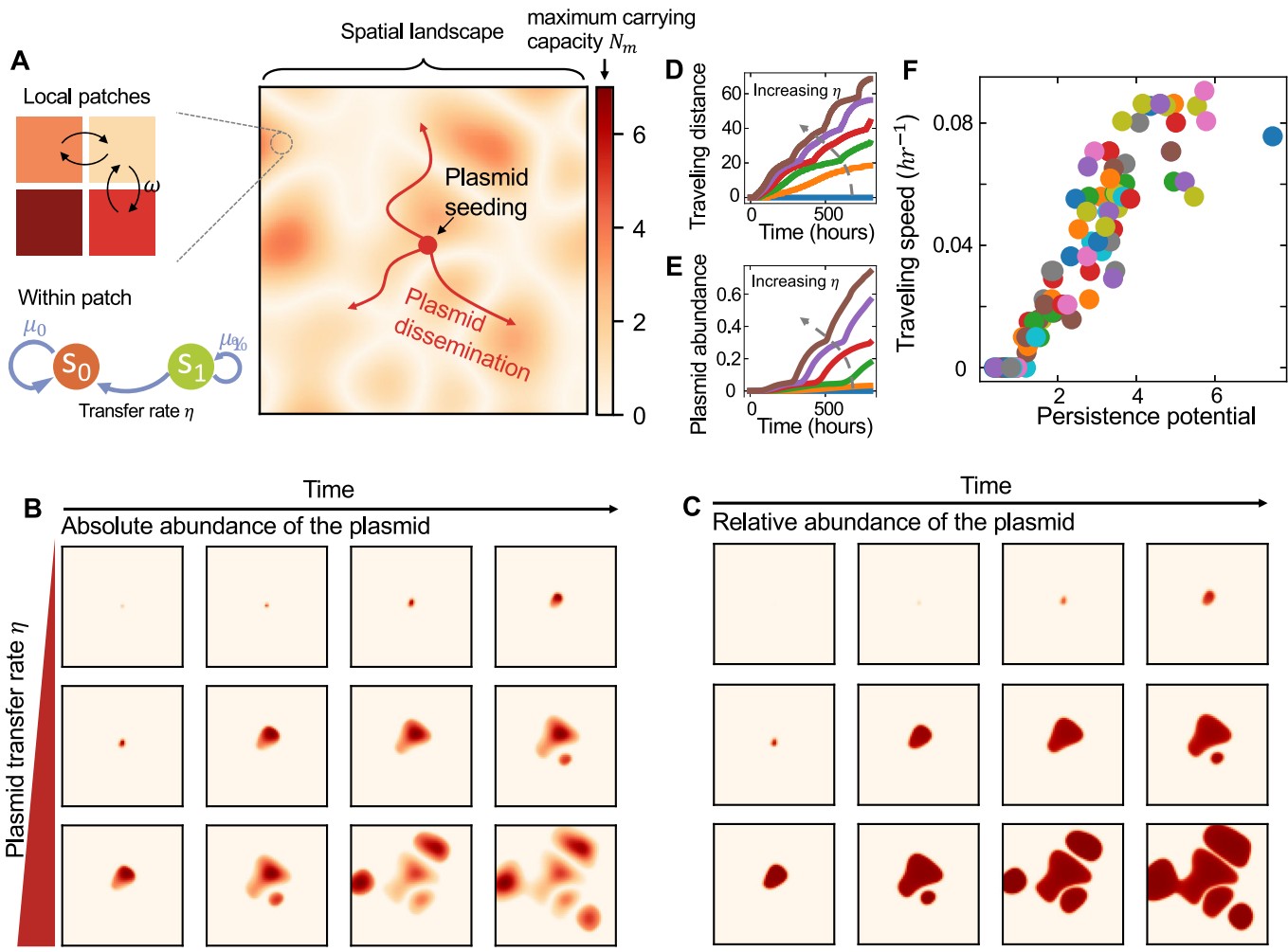

**Figure 2. A theoretical framework to simulate plasmid transfer dynamics in metacommunities composed of multiple patches.**

(A) The schematic of a metacommunity composed of 101 × 101 patches. The maximum carrying capacities (represented by color density) differed among patches, creating a spatial landscape where the plasmid traveled through. The plasmid was initially seeded at the central patch and disseminated towards peripheral patches via horizontal gene transfer. The within-patch dynamics is shaped by different processes: population growth by cell division, plasmid transfer from plasmid-carrying cells to plasmid-free ones, plasmid loss, and dilution. Here, $s_1$ and $s_0$ represent the local densities of cells carrying and not carrying the plasmid, respectively. $\mu_1$ and $\mu_0$ are their respective growth rates. $\eta$ and $\omega$ represent the within-patch and between-patch plasmid transfer rates, respectively. (B, C) Spatiotemporal dynamics of plasmid abundance in the metacommunity. The absolute and relative abundances (represented by color density) of the plasmid in each patch were calculated at different time points. Here, the relative abundance was calculated as the fraction of plasmid-carrying cells in each patch. Three different $\eta$ values (0.02, 0.05, and 0.1 h$^{-1}$) were tested and shown as examples. Other parameters were $\mu_0 = 0.5$ hr$^{-1}$, $\mu_1 = 0.45$ hr$^{-1}$, $\kappa = 0.01$ hr$^{-1}$, $D = 0.02$ hr$^{-1}$, $\omega = 0.04\eta$. The landscape of $N_m$ was randomly generated by Perlin noise generator. (D, E) Temporal dynamics of the plasmid's traveling distance and overall abundance in the metacommunity. Here, the traveling distance was calculated as the furthest straight-line distance that the plasmid had disseminated across from the seeding patch. The plasmid abundance was quantified as the fraction of plasmid-carrying cells in the total metacommunity. Six different $\eta$ values (0.01, 0.03, 0.05, 0.07, 0.09, and 0.11 h$^{-1}$) were tested and marked with different colors. Other parameters were $\mu_0 = 0.5$ hr$^{-1}$, $\mu_1 = 0.45$ hr$^{-1}$, $\kappa = 0.01$ hr$^{-1}$, $D = 0.02$ hr$^{-1}$, $\omega = 0.04\eta$. (F) The traveling speed of the plasmid is positively correlated with its persistence potential. The simulations were performed on a landscape created by Perlin noise generator. We randomized the parameters 100 times following uniform distributions in the ranges of $0.01 < \eta < 0.1$ hr$^{-1}$, $0.01 < \kappa < 0.02$ hr$^{-1}$, $0.3 < \mu_1 < 0.5$ hr$^{-1}$, $0.01 < D < 0.05$ hr$^{-1}$. Other parameters were $\mu_0 = 0.5$ hr$^{-1}$ and $\omega = 0.04\eta$. Persistence potential was calculated as $\eta/(D + \kappa - \frac{D}{1+\lambda})$.

event. Our results consistently showed that, in this system, reducing entropy promotes plasmid persistence and abundance (Appendix Fig. S6B–E). These results suggested a simple yet generalizable principle despite all the mechanistic complexities: reducing spatial entropy powered up the global persistence and dissemination of transferable plasmids.

Here we employed Perlin noise-like distributions to model the diverse spatial landscapes of metacommunities. However, the relationship between spatial entropy and plasmid maintenance is not dependent on any specific heterogeneity setups. Specifically, we tested additional heterogeneity setups generated using 2D Gaussian and 2D uniform distribution algorithms (Appendix Fig. S7A,B). We then simulated plasmid transfer dynamics within these metacommunities. Our results consistently showed that reducing spatial entropy enhances plasmid maintenance, regardless of the heterogeneity setup (Appendix Fig. S7C,D). This confirms that the

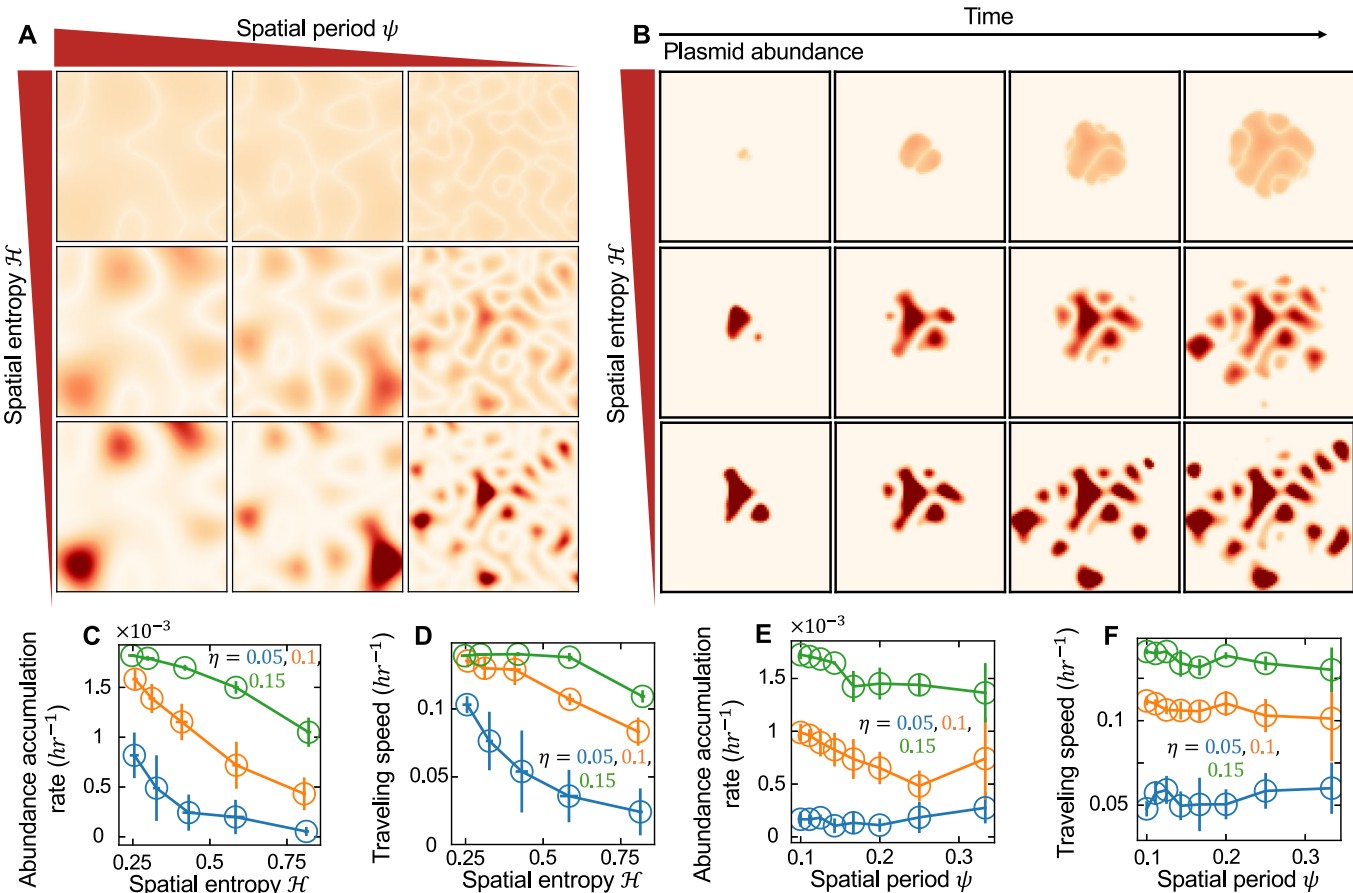

**Figure 3. Spatial entropy drives plasmid maintenance and dissemination.**

(A) Examples of landscapes with different spatial periods and entropies. The Perlin noise algorithm was used to generate these patterns. Three different octave numbers (2, 3, and 6) were tested from left to right. The data utilized to generate the middle column were identical to those employed in the middle column of Appendix Fig. S2A. (B) Spatiotemporal dynamics of plasmid abundance in metacommunities with different entropies. Three different entropies were tested from top to bottom. The absolute abundance (represented by color density) of the plasmid in each patch was calculated at different time points. Other parameters were $\mu_0 = 0.5\,hr^{-1}$, $\mu_1 = 0.45\,hr^{-1}$, $\kappa = 0.01\,hr^{-1}$, $D = 0.02\,hr^{-1}$, $\eta = 0.08\,hr^{-1}$, $\omega = 0.04\eta$. (C, D) Reducing spatial entropy sped up the accumulation and dissemination of the plasmid. Plasmid abundance accumulation rate and traveling speed were calculated by normalizing the total abundance and traveling distance at 200 h with time, respectively. Three different $\eta$ values were tested and shown as examples. Other parameters were $\mu_0 = 0.5\,hr^{-1}$, $\mu_1 = 0.48\,hr^{-1}$, $\kappa = 0.01\,hr^{-1}$, $D = 0.02\,hr^{-1}$, $\omega = 0.04\eta$. Data were presented as mean ± standard deviation of five replicates. Each replicate represented a unique spatial landscape generated by Perlin noise. (E, F) Spatial periodicity didn't significantly change the accumulation and dissemination of the plasmid. Periodicity was controlled by the octave parameter in the Perlin noise algorithm. Eight different octave values, from 3 to 10, were tested. Data were presented as mean ± standard deviation of five replicates.

effect of spatial entropy is independent of the specific spatial structure of the metacommunities.

Previous theoretical analysis in well-mixed populations predicted a threshold of horizontal gene transfer rate if a costly plasmid was to be maintained (Bergstrom et al, 2000; Lopatkin et al, 2017; Stewart and Levin, 1977; Wang and You, 2020). Below this critical rate, the plasmid transfer rate would be insufficient to overcome the fitness burden and segregation loss, and the plasmid would be predicted to be eliminated from the population (Fig. 4A,B). However, some studies showed that for many plasmids existing in nature, the predicted critical transfer rates might be too high to be biologically realistic (Bergstrom et al, 2000; Lili et al, 2007; Lopatkin et al, 2017; Simonsen, 1991; Sørensen et al, 2005). How to explain the stable persistence of many plasmids below the "theoretical limits" remained challenging (Lopatkin et al, 2017; Sørensen et al, 2005).

To test whether the theoretical limits predicted by the classic theories are still applicable in metacommunities with complex spatial landscapes, we calculated the steady-state abundance of the plasmid in the entire metacommunity as a function of transfer rate $\eta$ (Fig. 4B). Our results showed that there indeed existed a "theoretical limit" of the transfer rate in spatially homogeneous systems. Reducing spatial entropy, however, could significantly lower the critical transfer rate and allow the stable persistence of many plasmids below the "theoretical limit" (Fig. 4B). Given the ubiquity of spatial landscape in natural environments, our results provided an explanation for the stable existence of many selfish plasmids with slow transfer rates and highlighted the importance to consider spatial entropy when analyzing plasmid transfer dynamics.

The effects of spatial entropy on plasmid persistence and dissemination can be intuitively understood as follows: reducing

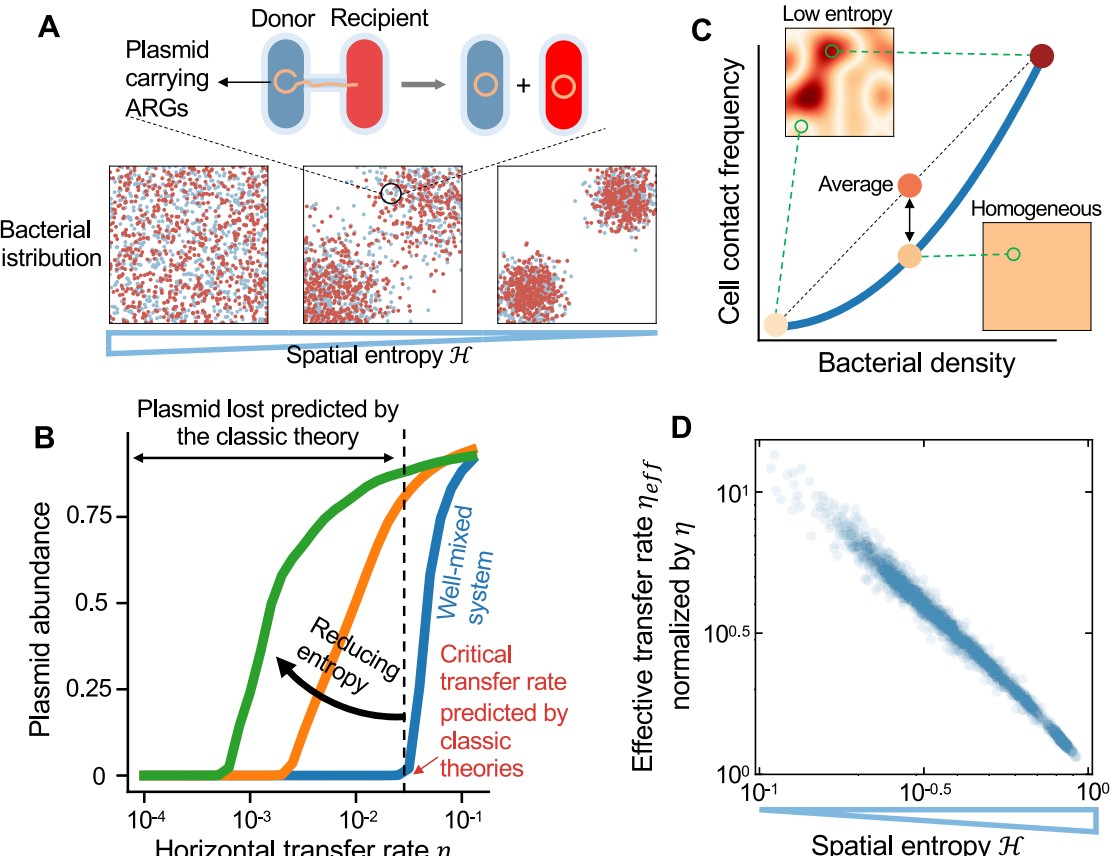

**Figure 4. Reducing spatial entropy allows the stable existence of many plasmids below the theoretical limits of plasmid transfer rates predicted by classic theories.**

(A) A schematic of plasmid transfer from donors (marked in blue) to recipients (marked in red) in bacterial populations with different spatial entropies. (B) The steady-state plasmid abundance in the metacommunity as a function of plasmid transfer rate $\eta$, under different spatial entropies. Three different entropy values (0.055, 0.216, and 1 from left to right, marked in green, yellow, and blue, respectively) were shown as examples. Here, numerical simulations were performed in metacommunities composed of 21 × 21 patches. The steady-state plasmid abundance was calculated as the fraction of plasmid-carrying cells in the metacommunity at 800 h. $\eta$ values ranging from $10^{-4}$ to $10^{-1}\,h^{-1}$ were tested. Other parameters were $\mu_0 = 0.5\,hr^{-1}$, $\mu_1 = 0.45\,hr^{-1}$, $\kappa = 0.005\,hr^{-1}$, $D = 0.2\,hr^{-1}$, $\omega = 0.04\eta$. (C) The cell contact frequency increases quadratically with the local cell density. Reducing spatial entropy causes the density fluctuations in the metacommunity, leading to the density decrease in some areas, while the increase in others. The gain of the contact frequency by density increase exceeds the loss resulting from local density reduction. Therefore, the average cell contact frequency gets elevated by entropy reduction. (D) The correlation between effective plasmid transfer rate $\eta_{eff}$ and spatial entropy. $\mathcal{H}$ equals 1 when the metacommunity is spatially homogeneous. Smaller $\mathcal{H}$ means greater heterogeneity of bacterial distributions.

entropy creates fluctuations of population densities in the space. Compared with the well-mixed populations, such fluctuations cause the density decrease in some areas, while the increase in others. Since the transfer of plasmids requires cell-to-cell contact, the transfer efficiency in general changes quadratically with the local cell density. The gain of the transfer efficiency by density increase exceeds the loss resulting from local density reduction (Fig. 4C; Appendix Fig. S8). Therefore, such fluctuations promote the average frequency of cell-to-cell contacts and facilitate the effective gene transfer rate in the space. Indeed, we simulated the reaction dynamics of plasmid transfer from donors to recipients in complex space, and our results suggested that the effective transfer rate ($\eta_{eff}$) was promoted by reducing spatial entropy $\mathcal{H}$ (Fig. 4D; see Method for details).

To experimentally validate the predicted relationship between $\eta_{eff}$ and $\mathcal{H}$, we constructed spatially heterogeneous *E. coli* metacommunities of two strains transferring a mobilizable plasmid. FM15 strain carrying F plasmid (conjugative, tetracycline resistant)

was used as donors, and MG1655 strain with chromosomal carbenicillin resistance served as recipients. After conjugation, the number of transconjugants could be quantified by Tet + Carb double selection. We allocated the mixture of donors and recipients into different groups of wells. Each group included ten wells and was treated as one metacommunity. We controlled the spatial entropy within each group by changing the cell density distributions among the wells (Fig. 5A; Appendix Table S1). The mean cell density across the ten wells was kept constant among different metacommunities, allowing us to distinguish the effect of spatial entropy from that of average density. After 1 h of incubation to allow plasmid transfer to take place, we calculated the effective plasmid transfer rate by averaging the entire metacommunity, including the low-density wells. Since we controlled the total amounts of donors or recipients in each group to be equal, the transconjugant number in each metacommunity reflected the effective transfer rate. Our results suggested that $\eta_{eff}$ was indeed promoted by reducing entropy (Fig. 5B). We further introduced

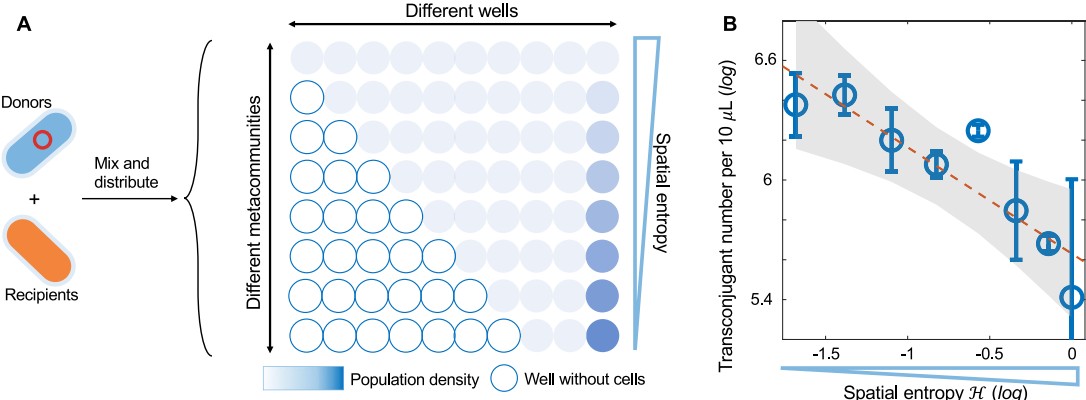

**Figure 5.  Experiments in metacommunities of *E. coli* strains transferring a mobilizable plasmid validated the predicted relationship between effective transfer rate and spatial entropy.**

(A) A schematic of the experimental design. Donor cells carrying the F plasmid were mixed with the recipients. The mixture was then allocated into different groups of wells. Each group contained the same amounts of cells, but the population distribution among the wells differed, creating a range of spatial entropies. After 1 h of incubation, the total number of transconjugants within each group was quantified by selective plating. (B) Reducing spatial entropy promoted the total number of transconjugants (per 10 μL) in each metacommunity. Here, since the total amounts of donors or recipients in different groups were equal, the transconjugant number in each group reflected the effective transfer rate $\eta_{eff}$. Data were presented as mean ± standard deviation of three replicates. The linear regression between log(transconjugant number) and log(spatial entropy) was shown as a dashed line. The shaded area represents a 95% confidence interval.

microbial dispersal into the experimental system by manually mixing the cell cultures of the neighboring wells. Every 15 min, 10 μL of bacterial culture was taken out from every well, transferred to and mixed with its adjacent wells (Appendix Fig. S9A). After one hour of incubation, the total number of transconjugants within each metacommunity was quantified using selective plating. Our results suggested that microbial dispersal created by manual mixing did not fundamentally change our conclusion: reducing spatial entropy still promoted the effective plasmid transfer rate in the metacommunities (Appendix Fig. S9B).

Our theory suggested that the decline of spatial entropy promoted the persistence and dissemination of plasmids. If this prediction holds true, prokaryotes from low-entropy environments will, in general, be expected to carry more plasmids compared with their counterparts from high-entropy environments. While the entropies in many natural habitats are difficult to directly measure, the compositions of functional genes in prokaryotic genomes might provide useful clues about their surrounding environments. For instance, prokaryotes carrying biofilm formation-related genes (BFGs) are more likely to live in biofilms, a typical low-entropy environment (Bostanghadiri et al, 2021; Kamali et al, 2020; Stewart and Franklin, 2008; Wimpenny et al, 2000; Zhuo et al, 2014). Therefore, BFGs can serve as an indicator of the environmental entropy, and our theory will predict the enrichment of plasmids in prokaryotes carrying BFGs.

To validate this prediction, we used NCBI RefSeq (Dataref: O'Leary et al, 2016), a comprehensive resource containing an extensive set of sequenced prokaryotic genomes (Fig. 6A; see Methods for details). We obtained a total of 34,688 complete genomes covering bacteria and archaea, 15,106 (43.55%) containing plasmids. All the genomes have been annotated following the PGAP pipeline (Tatusova et al, 2016), resulting in a large pool of 71,496 unique genes with known functions (see Methods for details). The large collection of annotated genomes allowed us to analyze the correlation between BFGs and plasmid carriage. We curated a comprehensive list of BFGs from the pool of functional genes, by matching the keyword 'biofilm' in the names of their protein products (Fig. 6A; Appendix Table S2). Based on this list, we identified 11,939 genomes (34.42%) that carried at least one BFG. Then we calculated the number of plasmids in genomes with or without BFGs. As shown in Fig. 6B,C, of all the genomes with BFGs, 63.25% contained at least one plasmid, with the mean plasmid number per cell being 1.77. In comparison, for genomes without BFGs, the plasmid-carriage fraction decreased to 33.21%, and the mean plasmid number per cell reduced to 0.82 (Appendix Fig. S10). The total plasmid size per genome exhibited the same trend (Fig. 6D), indicating that entropy reduction by biofilm formation promoted the maintenance of plasmids in prokaryotic genomes. We reached the same conclusion when focusing on the correlation between plasmid carriage and chromosomal BFGs (instead of BFGs in the entire genome) (Appendix Fig. S11).

Previous studies have suggested that genome size is positively associated with the number of MGEs (Khedkar et al, 2022; Newton and Bordenstein, 2011). This raises the possibility that the correlation between BFG carriage and plasmid abundance could be due to the larger genome sizes of prokaryotes carrying BFGs. We found that the genomes or chromosomes of prokaryotes with BFGs were indeed significantly larger than those without BFGs (Appendix Fig. S12A, B). To disentangle the role of spatial entropy from the influence of genome size, we divided the collection of prokaryotic genomes into multiple bins based on genome size. Within each bin, the size difference between the largest and smallest genomes was less than 2.4%, effectively minimizing the impact of genome size on plasmid carriage. We then analyzed whether the presence of BFGs led to a higher fraction of plasmid-carrying genomes in each bin. We found that in nearly all bins (13 out of 16), the fraction of plasmid-carrying genomes was higher among those with BFGs compared to those without BFGs

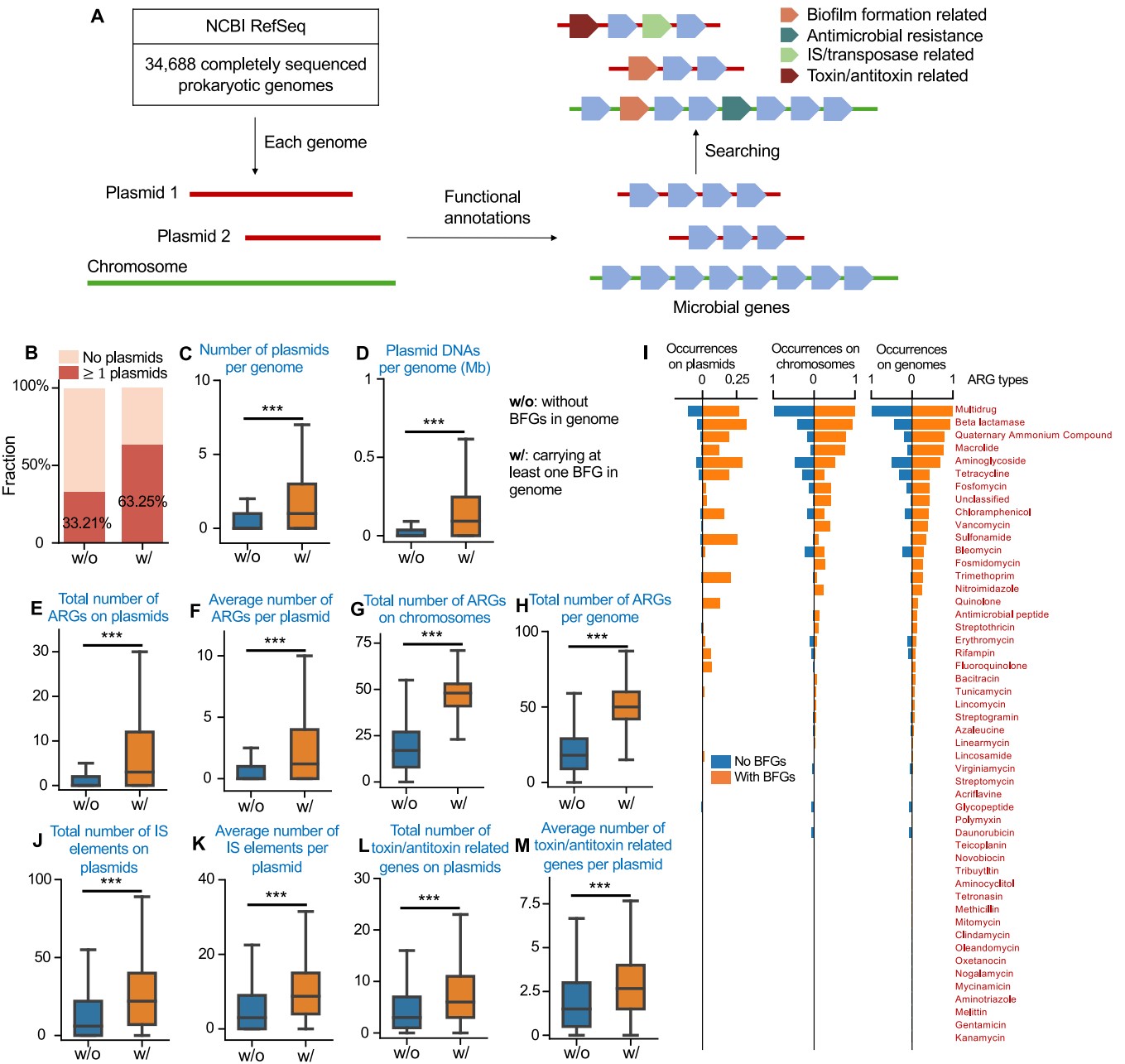

(Appendix Fig. S12C–E). These results suggest that the higher abundance of plasmids in the genomes with BFGs is not merely a consequence of their large genome sizes but is associated with low spatial entropy. More importantly, we found that the correlation between genome size and plasmid abundance only exists in genomes carrying BFGs. For genomes without BFGs, increasing genome size did not lead to an increase in plasmid number (Appendix Fig. S12C, D). These results indicate that the positive correlation between BFG presence and plasmid carriage can be independent of the effects of genome size.

We further compared the antibiotic resistance profiles of genomes with or without BFGs. To this end, we curated a comprehensive list of 2621 ARGs from the gene pool (see Methods for details). These ARGs belonged to 50 different resistance types, covering the commonly used antimicrobial drugs like beta-lactams, macrolides, aminoglycosides, and tetracycline (Appendix Fig. S13). Compared with plasmids from BFG-free genomes, the plasmids from BFG-carrying prokaryotes were equipped with significantly more abundant ARGs (Fig. 6E,F; Appendix Fig. S14A, B). The carriage of BFGs also brought about the enrichment of ARGs on a chromosome or the entire genome (Fig. 6G,H; Appendix Fig. S14C, D), probably by the intracellular transposition of ARGs between plasmids and the chromosome (Yao et al, 2022). These trends also held true for most separate types of ARGs (Fig. 6I). Collectively, these results supported our prediction that low-entropy environments promoted the maintenance and dissemination of antibiotic resistance.

Figure 6. Plasmids and ARGs are enriched in prokaryotic genomes carrying biofilm formation-related genes.

(A) A schematic of our analysis. From the NCBI RefSeq database, we retrieved 34,688 prokaryotic genomes that had been completely sequenced. 43.55% of the genomes carry at least one plasmid. The gene profile of each genome was determined by the PGAP annotation pipeline. Then, the biofilm formation-related genes (BFGs), antimicrobial resistance genes (ARGs), IS elements/transposases, and toxin/antitoxin-related genes were identified by searching the relevant keywords in the protein product names. (B) 33.21% of the genomes without biofilm-related genes carry plasmids. For genomes with biofilm-related genes, this fraction increases to 63.25%. (C) Genomes with BFGs ($n = 11,939$) in general carry a greater number of plasmids ($p < 0.001$) compared with those without BFGs ($n = 22,749$). For panels (C–H, J–M), the triple asterisk notation (***) denotes statistical significance at the $p < 0.001$ level, as determined by two-sided Student's $t$-tests. The box plots display the distribution of the data, where the central line represents the median (50th percentile), the box bounds (hinges) represent the 25th (Q1) and 75th (Q3) percentiles (interquartile range, IQR), and the whiskers extend to the minimum and maximum values within 1.5×IQR from Q1 and Q3, respectively. (D) Genomes with BFGs ($n = 11,939$) carry greater amounts of plasmid DNAs ($p = 2.22 \times 10^{-30}$) compared with those without BFGs ($n = 22,749$). Here, the amount of plasmid DNAs was calculated by summing the sizes of all different plasmids in the genome. (E) The genomes with BFGs ($n = 7552$) are equipped with a greater number of plasmid-borne ARGs ($p < 0.001$) than those without BFGs ($n = 7554$). Here, only the genomes carrying at least one plasmid were considered in the comparison. The total number of plasmid-borne ARGs was calculated by summing all the ARGs on each plasmid in the genome. (F) BFG carriage brings about the enrichment of ARGs in each plasmid ($p = 1.92 \times 10^{-135}$). Here, the average number of ARGs on each plasmid was calculated by normalizing the total number of ARGs by the plasmid number in the genome. Only the genomes carrying at least one plasmid were considered in this analysis ($n = 7554$ and 7552, respectively). (G) The abundances of chromosomal ARGs are higher ($p < 0.001$) in genomes with BFGs ($n = 11,939$) than in genomes without BFGs ($n = 22,749$). (H) The total number of ARGs per genome, calculated by summing up ARGs in the chromosomes and plasmids, is higher in genomes with BFGs ($p < 0.001$, $n = 22,749$ and 11,939 for the groups without and with BFGs, respectively). (I) BFG carriage is associated with a greater occurrence of different types of ARGs on plasmids, chromosomes, and genomes. Occurrence on plasmids were calculated as the proportion of the genomes that carried the type of ARGs on plasmids. Occurrence on chromosomes or genomes were calculated in a similar way. (J) The genomes with BFGs are equipped with a greater number of plasmid-borne IS elements ($p < 2.18 \times 10^{-63}$). Only the genomes carrying at least one plasmid were considered in the comparison ($n = 7554$ and 7552 for the groups without and with BFGs, respectively). (K) BFG carriage brings about the enrichment of IS elements in each plasmid ($p = 4.70 \times 10^{-66}$). The average number of IS elements on each plasmid was calculated by normalizing the total number of IS elements by plasmid number in the genome. Only the genomes carrying at least one plasmid were considered in the comparison ($n = 7554$ and 7552 for the groups without and with BFGs, respectively). (L) BFG carriage is also associated with the enrichment of toxin/antitoxin-related genes on plasmids ($p = 8.81 \times 10^{-74}$). Only the genomes carrying at least one plasmid were considered in the comparison ($n = 7554$ and 7552 for the groups without and with BFGs, respectively). (M) BFG carriage brings about the enrichment of toxin/antitoxin-related genes in each plasmid ($p = 4.38 \times 10^{-42}$, $n = 7554$ and 7552 for the groups without and with BFGs, respectively).

Insertion elements, also known as IS elements, typically containing a transposase gene flanked by inverted terminal DNA repeats, are another major type of MGEs (Siguier et al, 2006). By enabling the movement of ARGs into MGEs like plasmids, IS elements enhance the horizontal transfer of these genes between microbes (Yao et al, 2022). Our retrieved genomes contained a large collection of 3660 IS elements (see Methods for details). The carriage of BFGs significantly enriched the IS elements on plasmids and chromosomes (Fig. 6J,K; Appendix Fig. S15). These results suggested that the spread of IS elements might also be affected by spatial landscapes.

Toxins are the major pathogenicity factors produced by numerous bacteria (Jurėnas et al, 2022). By killing or inhibiting the growth of some co-resident strains, many bacterial toxins can also modulate the intraspecies interactions (Jurėnas et al, 2022; Popoff, 2020). A subset of toxins are paired with antitoxins that neutralize the toxic effects (Jurėnas et al, 2022). Toxin/antitoxin systems play major roles in stress response, plasmid maintenance, and bacterial persistence, and are commonly carried by MGEs (Jurėnas et al, 2022; Popoff, 2020). From the prokaryotic genome collection, we curated a list of 1635 toxin/antitoxin-related genes. These genes cover type I to V toxin/antitoxin systems and different toxin classes like endotoxins, enterotoxins, and bacteriocins (see Methods for details). The carriage of BFGs significantly promoted the abundance of toxin/antitoxin-related genes on prokaryotic chromosomes or plasmids (Fig. 6L,M; Appendix Fig. S16). These results highlight the role of spatial entropy in shaping microbial pathogenicity and species interactions.

To curate the list of BFGs, we employed a keyword-based approach, similar to methods used in previous biofilm-related studies. For example, Magalhães et al, conducted searches across various databases, including PubMed, using biofilm-related keywords to gather information on proteins and enzymes involved in biofilm formation (Magalhães et al, 2020). Zhang et al, employed text-mining analysis

with the keyword "biofilm" to identify biofilm-associated proteins (Zhang et al, 2024). To minimize the possibility that our approach might overlook any known biofilm-related genes, we conducted an extensive search using the NCBI Gene database (Brown et al, 2015), which provides detailed sequences, functional annotations, and related pathways of microbial genes. By filtering with the term "biofilm", we identified a total of 124 genes associated with biofilm formation (Appendix Table S3). Notably, 120 of these genes (96.8%) were already included in our initial list of BFGs (Appendix Table S2), indicating a high degree of coverage in our original analysis. To further expand our search scope, we employed additional filters, including "adhesion", "quorum sensing", "extracellular matrix", "exopolysaccharide biosynthesis", and "exopolysaccharide production" (Appendix Tables S4–S7). These terms are related to the process of biofilm formation. Using these filters, we generated lists containing 19, 39, 10, 27, and 9 genes, respectively. By integrating these newly identified genes with our previously recognized ones, we created an extended list of 240 non-redundant biofilm formation-related genes. To the best of our knowledge, this extended list encompasses most genes involved in biofilm formation, such as the *eps*, *psl*, and *sia* gene families (Chen et al, 2020; Marvasi et al, 2010; Overhage et al, 2005). We then repeated our analysis using this expanded list of BFGs. The results suggested that even with the inclusion of these additional genes, the presence of BFGs continued to promote the carriage and abundance of plasmids in prokaryotic genomes (Appendix Fig. S17). This finding underscores the robustness of our conclusions regarding the coverage of biofilm-related genes.

## Discussion

By combining mathematical modeling, experiments, and quantitative analysis on prokaryotic genome data, our results established a generalizable law governing the fates of plasmids in complex

environments. Despite the structural diversities of different microbial habitats, the maintenance and dissemination of a plasmid at broad scales are determined by a simple feature of the physical space, its entropy. Our work bridges the world of microbial ecology with one of the most fundamental concepts in physics and highlights the need for multidisciplinary perspectives when addressing the urgent challenges facing environments and human health.

The spatial pattern of microbial distribution can be characterized by two independent variables: the mean cell density over the entire space, and the evenness of density distribution. The effect of the first variable is well-known. In this work, we focused on the latter, which can be quantified by spatial entropy. In our simulations and experiments, we varied the entropy while maintaining a constant average density across the metacommunities. Our results consistently showed that, regardless of the mean microbial density, reducing entropy (or promoting unevenness) enhances the survival and dissemination of plasmids within the metacommunities.

Alternative explanations exist for the observed correlation between the presence of BFGs and plasmid abundance. For instance, biofilm-forming microbes might have higher fitness, enabling them to maintain more plasmids. Biofilms can indeed provide several advantages that enhance microbial fitness, such as physical protection against environmental stresses and optimized resource acquisition (Flemming et al, 2016). However, the relative fitness of biofilm-dwelling microbes compared to their free-living counterparts can vary significantly depending on the microbial species and environmental conditions (Flemming et al, 2016). For example, in nutrient-scarce environments, biofilm microbes may have less access to resources than free-living cells that can move more freely. Similarly, in low-oxygen environments, aerobic microbes in biofilms may be disadvantaged compared to free-living cells that can access oxygen more readily. Additionally, biofilms can accumulate toxins or waste products that are harmful to the microbes within them (Bonnineau et al, 2021). Given these complex contexts, genomic sequences alone may be insufficient to reliably estimate the fitness of microbes living in or out of biofilms. Disentangling the effects of microbial fitness may require more in-depth quantification of microbial growth dynamics in real environments.

Plasmids can carry genes that promote microbial clustering. To evaluate the contributions of plasmids to biofilm formation, we calculated the number of BFGs carried by each plasmid. Among the 34,688 genomes we analyzed, 11,939 (34.4%) carried at least one BFG. However, only a small fraction (1.8%) of these genomes had BFGs located on plasmids. In total, we identified 71,522 BFGs across all genomes, with plasmids contributing only 278 genes (~0.39%). Thus, most genes promoting microbial clustering are chromosomally encoded, and the contribution of plasmids to biofilm formation appears to be relatively minor (Appendix Fig. S18). To further examine the influence of plasmid-borne BFGs on our conclusions, we repeated our analysis after excluding genomes containing BFGs on plasmids. Our results still indicated that biofilm formation promotes the maintenance of plasmids in prokaryotic genomes, even when plasmids do not contribute to biofilm formation (Appendix Fig. S19). These findings suggest that the observed correlation between biofilm formation capability and plasmid abundance is highly likely to be mediated by the entropy effects rather than direct contributions from plasmids.

Our work predicts that many non-selective factors in natural environments can contribute to the dissemination of ARGs by reducing the spatial entropy of the population. This prediction is in line with empirical observations in different contexts. For instance, previous studies suggested that microplastics promoted the abundance of ARGs in soil or aquatic microbiota (Lin et al, 2024; Zhu et al, 2022). It has also been shown that the airborne ARGs on inhalable particles were enriched in hazy days compared with non-hazy days, indicating an association between air pollution and ARG dissemination (Sun et al, 2020). Discarded facial masks, when incubated in an estuary, increased the abundance of ARGs, potentially by reshaping the spatial distribution of bacteria in the surrounding aquatic environment (Lin et al, 2022). This insight can be valuable for the management and control of antibiotic resistance: Reducing antibiotic exposure might be insufficient to reverse the ARG persistence (Lopatkin et al, 2017; Martínez et al, 2015; Zhang et al, 2022). Instead, disrupting the low-entropy condition can be the key. In soil or water, spatial entropy of microbiota can be promoted by removing plastic pollution. In clinical settings, entropy can be elevated by disrupting biofilms using dispersing enzymes (Wang et al, 2023), nanoparticles (Ding et al, 2024; Ma et al, 2022) or monoclonal antibody (Kurbatfinski et al, 2023). Our work suggested that these approaches might promote the efficacy of ARG clearance.

Given the diversity of the genes carried by plasmids, the effects of spatial landscape on microbial functions should not be limited to antibiotic resistance. Indeed, plasmids also encode a wide variety of metabolic capabilities or adaptive traits (Rankin et al, 2011). Examples include nitrogen fixation (Nuti et al, 1979), photosynthesis (Paul and Sullivan, 2005), amino acid production (Gil et al, 2006), herbicide degradation (Kellogg et al, 1981), utilization of uncommon nutrients (Heuer and Smalla, 2012) or heavy metal tolerance (Silver and Misra, 1988). The abundance of plasmids in a prokaryote shapes its adaptability and genome plasticity (Frost et al, 2005). Plasmids also drive bacterial social behaviors, by producing secreted proteins that are mutually beneficial or spiteful for other individuals (Rankin et al, 2011). Thus, we anticipate that the spatial landscape of the physical environment can be a global regulator of microbial functionality, sociality, and evolvability. We believe such insights can provide new opportunities for precision microbiome engineering.

# Methods

**Reagents and tools table**

| Reagent/resource | Reference or source | Identifier or catalog number |
| --- | --- | --- |
| **Experimental models** | | |
| *E. coli* FM15 strain with F plasmid | Fu lab | N/A |
| *E. coli* MG1655 | Fu lab | N/A |
| **Recombinant DNA** | | |
| **Antibodies** | | |
| **Oligonucleotides and other sequence-based reagents** | | |
| **Chemicals, enzymes, and other reagents** | | |
| LB Broth | HuanKai Microbial | 028320 |
| Tetracycline | Macklin | C14975265 |
| Carbenicillin | Solarbio | 1129Z033 |

| Reagent/resource | Reference or source | Identifier or catalog number |
|---|---|---|
| **Software** | | |
| Visual Studio Code | https://code.visualstudio.com/ | N/A |
| Perlin noise 1.13 | https://pypi.org/project/perlin-noise/ | N/A |
| **Other** | | |

## Theoretical framework and numerical simulations of plasmid transfer dynamics in metacommunities

Consider the dissemination of one plasmid in a metacommunity of $m \times m$ patches occupied by a single species. We assumed that within each patch, the population was well-mixed. The system dynamics in each patch can be described by two ordinary differential equations (ODEs):

$$\frac{ds_0}{dt} = \mu_0 s_0 \left( 1 - \frac{s_0 + s_1}{N_m} \right) - \eta s_0 s_1 + \kappa s_1 - D s_0 - \omega s_0 S_{adj},$$

$$\frac{ds_1}{dt} = \mu_1 s_1 \left( 1 - \frac{s_0 + s_1}{N_m} \right) + \eta s_0 s_1 - \kappa s_1 - D s_1 + \omega s_0 S_{adj}.$$

Here, $s_1$ and $s_0$ represent the local densities of cells carrying and not carrying the plasmid, respectively. $\mu_1$ and $\mu_0$ are their respective growth rates. $\mu_1$ is related to $\mu_0$ via $\mu_1 = \frac{\mu_0}{1+\lambda}$, where $\lambda$ stands for the fitness effect of the plasmid. Positive $\lambda$ represents fitness burden. $N_m$ is the maximum carrying capacity of the patch. $N_m$ varies across different patches, and the distribution of $N_m$ value defines the spatial landscape of the metacommunity. $\eta$ is the within-patch plasmid transfer rate from the donors to recipients. $\kappa$ is the plasmid loss rate. $D$ is the dilution rate. $\omega$ stands for the cross-patch transfer rate of the plasmid. $S_{adj}$ is the sum of the donor densities in all the adjacent patches.

Without loss of generality, we constructed metacommunities of $101 \times 101$ patches. Each patch was initially fully colonized by plasmid-free cells. Then we seeded the plasmid-carrying cells in the central patch, with their initial fraction in the patch being 1%. The cross-patch plasmid transfer allowed the plasmid to disseminate towards the peripheral patches. Then we simulated the temporal dynamics of plasmid abundance in each patch, which allowed us to analyze the traveling speed (denoted as $v$) of the plasmid and its abundance accumulation rate (denoted as $u$) in the entire space.

When calculating $v$, we focused on the furthest straight-line distance that the plasmid had traveled across from the seeding point. We applied a threshold of 0.01 on the relative abundance of the plasmid in each patch. Only when the relative abundance exceeded this threshold, the plasmid was treated as "present" in the patch. At each timepoint, we calculated the number of patches where the plasmid was present. We also calculated the distances of these patches from the seeding point. The maximum distance among them was used as the furthest straight-line distance that the plasmid had traveled across. Then the traveling speed was quantified as the average speed in the first 200 h. The abundance accumulation rate was calculated in a similar way.

## Simulating the spatial dissemination of plasmids with different persistence potentials

Previous studies suggested that the abundance of a plasmid in a well-mixed population could be predicted by its persistence potential (Lopatkin et al, 2017; Wang and You, 2020), which could be approximately calculated as $\eta/(D + \kappa - \frac{D}{1+\lambda})$. Only when its persistence potential is larger than 1, the plasmid can be stably maintained in the population. Otherwise, the plasmid-carrying cells will be outcompeted by the plasmid-free cells. To understand the role of persistence potential on plasmid dissemination, we performed numerical simulations with randomized parameters. In particular, we randomized the parameters 100 times following uniform distributions in the ranges of $0.01 < \eta < 0.1\,\text{hr}^{-1}$, $0.01 < \kappa < 0.02\,\text{hr}^{-1}$, $0.3 < \mu_1 < 0.5\,\text{hr}^{-1}$, $0.01 < D < 0.05\,\text{hr}^{-1}$. For each combination of parameters, we simulated the plasmid transfer dynamics on a given landscape. Then we calculated the dissemination speed and abundance accumulation rate as functions of the persistence potential.

## Generation of random spatial landscapes using Perlin noise

Perlin noise is a powerful algorithm to generate randomized landscapes. The mathematics and logic of the algorithm were introduced in detail in previous works (Etherington, 2022; Perlin, 1985). The first step is to define a 2-dimensional grid where each intersection in a rectangle is associated with a random unit-length gradient vector. Then, for each point in a grid cell, four offset vectors are defined, each connecting a corner to the point. For each corner, the dot product between its gradient vector and the offset vector is calculated. The final step is interpolation between the four dot products, which is performed using a function with zero first derivative at the grid nodes. The interpolation step will give Perlin noise its characteristic look.

In our numerical simulations, Perlin noise was implemented using the "perlin-noise" package (v1.12, released: Jan 29, 2022, from: https://pypi.org/project/perlin-noise/) in Python. This algorithm contains two parameters: octave, which controls the number of subrectangles in each [0, 1] range, and seed number, which specifies the seed to initialize the random number generator. The octave parameter determines the periodicity of the pattern. Larger octave leads to a greater frequency of the recurring structures. Therefore, we quantified the spatial period (denoted as $\psi$) of a generated landscape using the reciprocal of the octave number. For a given number ($m$) of patches on each of the two dimensions, the algorithm returns a $m \times m$ matrix $[a_{ij}]$ ($1 \leq i \leq m, 1 \leq j \leq m$). We calculated the maximum carrying capacity $N_m$ at the patch $[i, j]$ as $z \frac{|a_{ij}|^n}{\sum_{i,j} |a_{ij}|^n} \cdot m^2$, such that the mean value of $N_m$'s in all the patches equaled $z$. The mean cell density in the metacommunity can be controlled by changing $z$. $|a_{ij}|$ is the absolute value of $a_{ij}$ and $n$ is an integer. The value of $n$ controls the fluctuation magnitude of bacterial density across space. Larger $n$ leads to smaller spatial entropy. When $n$ equals zero, all $N_m$ values become identical, leading to a homogeneous distribution of microbes in the metacommunities.

## Calculating the spatial entropy of a randomly generated landscape

For a metacommunity of $m \times m$ patches, the maximum carrying capacity $N_m$ at the patch $[i, j]$ was $z \frac{|a_{ij}|^n}{\sum_{i,j}|a_{ij}|^n} \cdot m^2$. Let $x_{ij}$ be $\frac{|a_{ij}|^n}{\sum_{i,j}|a_{ij}|^n}$. The spatial entropy of the landscape was calculated as $\mathcal{H} = \frac{\exp\left(-\sum_{i,j} x_{ij} \log x_{ij}\right)}{m^2}$. $\mathcal{H}$ equals 1 when the metacommunity is spatially homogeneous. Smaller entropy means greater heterogeneity of bacterial distributions.

## Quantification of effective transfer rate $\eta_{eff}$ in complex space

For a metacommunity composed of $n$ patches, let $x_i$ be the local density of donors and recipients in the $i$-th patch and $p_i$ be $\frac{x_i}{\sum_{i=1}^{n} x_i}$. Here, we assumed that donors and recipients shared the same spatial distribution. Let $V_i$ be the volume of the $i$-th patch and $V$ be the total volume of all patches. Without loss of generality, we assumed that each patch has the same volume. Within the time period $\Delta t$, the density gain of plasmid-carrying cells in the $i$-th patch could be obtained as $\eta x_i^2 \Delta t$. Therefore, the number of plasmid-carrying cells in the metacommunity increased by $\eta \Delta t V_i \sum_{i=1}^{n} x_i^2$. The effective transfer rate of the plasmid in the entire space could then be calculated as:

$$\eta_{eff} = \frac{(\eta \Delta t V_i \sum_{i=1}^{n} x_i^2)/V}{(\sum_{i=1}^{n} x_i V_i)/V \cdot (\sum_{i=1}^{n} x_i V_i)/V} \cdot \frac{1}{\Delta t}$$

which led to $\eta_{eff}/\eta = n \cdot \sum_{i=1}^{n} p_i^2$.

The spatial entropy of this metacommunity was quantified as $\mathcal{H} = \frac{\exp\left(-\sum_{i=1}^{n} p_i \log p_i\right)}{n}$. We performed numerical simulations to examine the relationship between entropy $\mathcal{H}$ and $\eta_{eff}/\eta$. In particular, we randomized $n$ between 5 and 100. For each $n$, we created a random distribution of $x_i$ between 0 and 1, which then allowed us to calculate the values of $\mathcal{H}$ and $\eta_{eff}/\eta$.

## Experimental validation of the effects of spatial entropy

In experiments, we constructed spatially heterogeneous metacommunities using *E. coli* strains transferring a mobilizable plasmid. FM15 strain carrying F plasmid (conjugative, tetracycline resistant) was used as donor, and MG1655 strain with chromosomal carbenicillin resistance was used as recipient. Upon conjugation, the number of transconjugants can be measured by Tet + Carb double selections (20 and 50 µg/ml, respectively). Sixteen hours overnight cultures of donors and recipients (5 mL LB media with appropriate selecting agents) were collected, washed twice (5000 rpm, 3 min) using fresh LB media to remove the selecting agents, and resuspended (fresh LB media). The donor population was mixed with the recipient population. The mixture was then immediately allocated into different groups of wells (200 µL for each well). Each group included 10 wells, and we controlled the spatial entropy within the group by changing the cell density distributions among the wells, as shown in Appendix Table S1. We did not employ randomization procedures for sample allocation, nor did we implement blinding during experiments or data analysis. After 1 h of incubation (room temperature, in the absence of shaking), we mixed the populations in ten wells together. The mixture of each group was diluted 0.1 fold, and 10 µL of the diluted mixture was plated on Tet + Carb double selective agar plates. After 24 h of incubation at 37 °C we counted the number of transconjugants on each plate.

## Collection of annotated prokaryotic genomes

The NCBI RefSeq database provides up-to-date and consistent annotations for a large collection of bacterial and archaeal genomes (Haft et al, 2024; O'Leary et al, 2016). The annotations were carried out using Prokaryotic Genome Annotation Pipeline (PGAP), which predicted genes and other functional elements based on sequence (Tatusova et al, 2016). We downloaded a total of 34,688 completely sequenced and annotated prokaryotic genomes from the database. The replicons in each genome were labeled as chromosomes or plasmids. From each chromosome or plasmid, we developed custom codes to extract its size and all the protein products that the replicon encoded based on the annotated genome profile. A total of 71,496 protein products were obtained from all the genomes.

## Curation of BFGs, ARGs, IS elements, and toxin/antitoxin-related genes

The KEGG database maintains a comprehensive list of antimicrobial resistance genes (https://www.genome.jp/brite/ko01504). By matching the names of protein products with this list, we identified 2621 antibiotic resistance genes encoded by chromosomes or plasmids. Meanwhile, we categorized the resistance genes into 50 different classes, based on the resistance types. We identified 137 BFGs from all the protein products by searching through their names using 'biofilm' as the keyword. Similarly, we identified 1635 toxin/antitoxin-related genes by matching the keywords "toxin", "hemolysin", "bacteriocin", "colicin", "cytolysin", "pyocin", "hemagglutinin", "leukocidin", "streptolysin", "superantigen", and "hydrogen cyanide synthase". Additionally, we identified 3660 IS elements-related genes by matching the keywords "transposase", "insertion element", "IS-like element", "IS-family protein", and "ICE mobile element".

# Data availability

The datasets and computer code produced in this study are available in the following databases: Source data and modeling computer scripts: GitHub (https://github.com/twang1993/SpatialEntropy).

The source data of this paper are collected in the following database record: biostudies:S-SCDT-10_1038-S44320-025-00110-8.

# Peer review information

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

## Acknowledgements

This study was supported by the National Key R&D Program of China (2024YFA0920200 to TW), the National Natural Science Foundation of China (32470701 and 12401660 to TW), and the Shenzhen Institute of Synthetic Biology Scientific Research Program (HSE499011086 to TW). We are grateful to the Shenzhen Infrastructure for Synthetic Biology for providing instrument support and technical assistance.

## Author contributions

**Wenzhi Xue**: Data curation; Formal analysis; Validation; Investigation; Methodology; Writing—original draft. **Juken Hong**: Formal analysis; Validation; Investigation; Methodology; Writing—original draft; Modeling. **Runmeng Zhao**: Modeling. **Huaxiong Yao**: Validation. **Yi Zhang**: Resources. **Zhuojun Dai**: Resources; Validation. **Teng Wang**: Conceptualization; Resources; Formal analysis; Supervision; Funding acquisition; Validation; Investigation; Methodology; Writing—original draft; Writing—review and editing; Modeling.

The source data underlying the figure panels in this paper may have individual authorship assigned. Where available, figure panel/source data authorship is listed in the following database record: biostudies:S-SCDT-10_1038-S44320-025-00110-8.

## Disclosure and competing interests statement

The authors declare no competing interests.

