## [Peer Review File · Molecular Systems Biology]

Spatial entropy drives the maintenance and dissemination of transferable plasmids

Wenzhi Xue, Juken Hong, Runmeng Zhao, Huaxiong Yao, Yi Zhang, Zhuojun Dai, and Teng Wang

Corresponding author(s): Teng Wang (t.wang1@siat.ac.cn)

Review Timeline:

Submission Date:	21st Oct 24
Editorial Decision:	27th Nov 24
Revision Received:	10th Feb 25
Editorial Decision:	26th Mar 25
Revision Received:	31st Mar 25
Accepted:	11th Apr 25

Editor: Poonam Bheda

Transaction Report:

27th Nov 2024

Manuscript Number: MSB-2024-12666

Title: Spatial entropy drives the maintenance and dissemination of mobile genetic elements

Dear Prof Wang,

Thank you for the submission of your manuscript to Molecular Systems Biology. We have now received feedback from the three reviewers who agreed to evaluate your manuscript. As you will see from the reports below, the referees acknowledge the interest of the study and are overall supportive of your work; however they also comment on multiple aspects of the manuscript that should be strengthened in a revision.

I think that the recommendations of the reviewers are rather clear and I therefore do not see the need to repeat the comments listed below. Two of the more fundamental points raised refer to whether the well assay can indeed recapitulate community interactions and that potential issues with the mathematical model assumptions need to be further examined.

All other issues raised would need to be satisfactorily addressed. Please let me know in case you would like to discuss in further detail any of the issues raised, I would be happy to schedule a call.

We require:

1) A .docx formatted version of the manuscript text (including legends for main figures, EV figures and tables). Please make sure that the changes are highlighted to be clearly visible. Alternatively you may choose to submit your manuscript as a LaTeX file.

4) A .docx formatted letter INCLUDING the reviewers' reports and your detailed point-by-point responses to their comments. As part of the EMBO Press transparent editorial process, the point-by-point response is part of the Peer Review File (PRF), which will be published alongside your paper.

5) A complete author checklist, which you can download from our author guidelines (<https://www.embopress.org/page/journal/17574684/authorguide#submissionofrevisions>). Please insert information in the checklist that is also reflected in the manuscript. The completed author checklist will also be part of the PRF.

6) Please note that all corresponding authors are required to supply an ORCID ID for their name upon submission of a revised manuscript.

7) It is mandatory to include a 'Data Availability' section after the Materials and Methods. Before submitting your revision, primary datasets produced in this study need to be deposited in an appropriate public database, and the accession numbers and database listed under 'Data Availability'. Please remember to provide a reviewer password if the datasets are not yet public (see <https://www.embopress.org/page/journal/17574684/authorguide#dataavailability>).

This study includes no data deposited in external repositories.

8) All Materials and Methods need to be described in the main text using our 'Structured Methods' format, which is required for all research articles. According to this format, the Methods section includes a Reagents and Tools Table (listing key reagents, experimental models, software and relevant equipment and including their sources and relevant identifiers) followed by a Methods and Protocols section describing the methods using a step-by-step protocol format. The aim is to facilitate adoption of the methodologies across labs. Please upload the Reagents and Tools table as a separate document when submitting your revised manuscript. More information on how to adhere to this format as well as a downloadable template (.docx) for the Reagents and Tools Table can be found in our author guidelines:

<https://www.embopress.org/page/journal/17444292/authorguide#structuredmethods>

An example of a Method paper with Structured Methods can be found here:
<https://www.embopress.org/doi/10.15252/msb.20178071>.

9) For data quantification: please specify the name of the statistical test used to generate error bars and P values, the number (n) of independent experiments (specify technical or biological replicates) underlying each data point and the test used to calculate p-values in each figure legend. The figure legends should contain a basic description of n, P and the test applied. Graphs must include a description of the bars and the error bars (s.d., s.e.m.). Please provide exact p values.

10) Our journal encourages inclusion of *data citations in the reference list* to directly cite datasets that were re-used and obtained from public databases. Data citations in the article text are distinct from normal bibliographical citations and should directly link to the database records from which the data can be accessed. In the main text, data citations are formatted as follows: "Data ref: Smith et al, 2001" or "Data ref: NCBI Sequence Read Archive PRJNA342805, 2017". In the Reference list, data citations must be labeled with "[DATASET]". A data reference must provide the database name, accession number/identifiers and a resolvable link to the landing page from which the data can be accessed at the end of the reference. Further instructions are available at .

11) We replaced Supplementary Information with Expanded View (EV) Figures and Tables that are collapsible/expandable online. A maximum of 5 EV Figures can be typeset. EV Figures should be cited as 'Figure EV1, Figure EV2' etc... in the text and their respective legends should be included in the main text after the legends of regular figures.

<https://www.embopress.org/page/journal/17574684/authorguide#expandedview>

13) Author contributions: CRedit has replaced the traditional author contributions section because it offers a systematic machine readable author contributions format that allows for more effective research assessment. Please remove the Authors Contributions from the manuscript and use the free text boxes beneath each contributing author's name in our system to add specific details on the author's contribution. More information is available in our guide to authors.

Please also suggest a striking image or visual abstract to illustrate your article as a PNG file 550 px wide x 300-600 px high. Share synopsis text and image, as well as eTOC:

Please note that these would be the final versions and changes during proofing are usually not allowed

16) As part of the EMBO Publications transparent editorial process initiative (see our policy here:

https://www.embopress.org/transparent-process#Review_Process), Molecular Systems Biology will publish online a Peer Review File (PRF) to accompany accepted manuscripts.

In the event of acceptance, this file will be published in conjunction with your paper and will include the anonymous referee reports, your point-by-point response and all pertinent correspondence relating to the manuscript. Let us know whether you agree with the publication of the PRF and as here, if you want to remove or not any figures from it prior to publication.

Please note that the Authors checklist will be published at the end of the PRF.

Molecular Systems Biology has a "scooping protection" policy, whereby similar findings that are published by others during review or revision are not a criterion for rejection. Should you decide to submit a revised version, I do ask that you get in touch after three months if you have not completed it, to update us on the status.

I look forward to receiving your revised manuscript.

Yours sincerely,

Poonam Bheda, PhD
Scientific Editor
Molecular Systems Biology

Reviewer #1:

The manuscript "Spatial entropy drives the maintenance and dissemination of mobile genetic elements" attempts to address a knowledge gap in the field concerning the influence of spatial landscape in maintenance and spread of mobile elements in environmental microbial communities. This study is important, timely and of significance as most of our current knowledge about MGE dynamics stems from studying bacteria and their MGEs in well-mixed populations, but these do not represent the spatial landscape in complex environmental communities.

The authors use a sound theoretical framework to represent microbial metacommunities in the environment and simulate gene flow dynamics. Based on the simulations where they keep cell densities constant while varying spatial entropy, the authors draw their main conclusion that lowering of spatial entropy favors maintenance and spread of mobile elements (MGE). The simulation methodology is rigorous and results based on them are convincing. The authors make a conceptual advance and also provide necessary context to explain their observations. They attribute their main conclusion to - resulting high local cell densities in low entropy environments, and establish relationship between spatial entropy and average rate of MGE transfer. They also attempt to validate this relationship based in theory using (1) an experimental set-up that measures MGE spread in *E. coli* metacommunities, and (2) through genome sequence analysis of bacteria from high and low spatial entropy environments. However, the manuscript falls short in providing convincing validation through their experimental and quantitative genomic analysis. Hence, the manuscript can be revised based on following suggestions.

Major points -

1. Based on the description of *E. coli* metacommunities experiment the wells in each row represent a single meta-community. However, these wells are not connected i.e, a physical barrier separates members of the metacommunity. Given this, I would argue that the experiment does not represent the spatial landscape from the simulations described in Lines 82-85 - where the local patches are not separated by a barrier and allow for bacterial dispersal or immigration. Again, in the introduction Lines 51-53 the authors state that "the global maintenance of MGEs also depends on its efficiency to disseminate across different areas", however the experimental set-up does not support this condition. Hence, in my opinion the current experimental set-up with physical barriers is insufficient to validate the conditions used in the simulations.

2. In support of their theory, the authors test the expectation that bacteria in low-entropy environments like biofilms should carry more MGEs compared to high entropy environments as low spatial entropy promotes MGE persistence and spread. To address this, they perform genome sequence analysis and stratify bacteria living in low entropy and high entropy environments based on presence or absence respectively of biofilm formation related genes (BFGs). The authors find that bacteria with BFGs carry more plasmids compared to bacteria without BFGs. However, genome size is positively correlated with the number of MGEs(1,2). Hence, it is important to first determine that bacteria with BFGs do not have significantly larger genomes compared to bacteria without BFGs. This analysis is important to convincingly refute that the higher fraction of MGEs in the genomes with BFGs is not merely a consequence of their large genome sizes but is associated with low spatial entropy.

Minor points -

1. Throughout the manuscript both in the simulations and experiments - the authors refer to the umbrella term of mobile elements or MGEs but the assumptions, environmental conditions and even the mechanisms specifically refer to MGEs that transfer via cell-to-cell contact, a particular MGE type i.e, plasmids. Hence, it might benefit to avoid generalisations and refer to mobile elements as plasmids throughout the manuscript.
2. In Figure 2a the scale requires a legend.
3. Description of Figure 2a needs text describing within patch dynamics and description for the terms S_0 , S_1 , 0, 1
4. For Figure 4d and 5b, to improve readability the respective ends of x - axis could be marked high and low alongside their marked spatial entropy values. In addition, Line 450 from methods section could be added to the description text of Figure 4d for

ease of understanding.

References:

1. I. L. G. Newton, S. R. Bordenstein, Correlations Between Bacterial Ecology and Mobile DNA. *Curr. Microbiol.* 62, 198-208 (2011).
2. S. Khedkar, G. Smyshlyaev, I. Letunic, O. M. Maistrenko, L. P. Coelho, A. Orakov, S. K. Forslund, F. Hildebrand, M. Luetge, T. S. B. Schmidt, O. Barabas, P. Bork, Landscape of mobile genetic elements and their antibiotic resistance cargo in prokaryotic genomes. *Nucleic Acids Res.* 50, 3155-3168 (2022).

Reviewer #2:

This manuscript "Spatial entropy drives the maintenance and dissemination of mobile genetic elements" addresses fundamental questions regarding how spatial heterogeneity influences the maintenance and spread of mobile genetic elements (MGEs) in microbial communities. The authors present a novel theoretical framework that links spatial entropy to MGE persistence, supported by experimental validation and large-scale genomic analysis. This manuscript makes a significant contribution to understanding MGE maintenance and spread in spatially structured environments. , and is particularly relevant given the growing concerns about antibiotic resistance and its spread through MGEs.

The study effectively bridges theoretical predictions with experimental validation and genomic evidence, and the introduction of spatial entropy as a unified measure for analyzing MGE persistence is innovative, providing implications for understanding and potentially controlling antibiotic resistance spread

However, there are some issues to be addressed, first is about the mathematical model assumptions. I think the relationship between density and transfer efficiency requires more thorough examination. The assumption that "gain of transfer efficiency by density increase exceeds the loss from local density reduction" may impose specific requirements on the "density-transfer efficiency" relationship. For example, the near-exponential increase in density-bacterial contact frequency (Figure 4c) is crucial for the results. What happens if the relationship saturates? Could density increases in certain regions fail to compensate for decreased transfer rates elsewhere? While the general conclusion that "spatial heterogeneity promotes coexistence" aligns with previous ecological theories, testing different functional forms would strengthen the mathematical foundation

Also, about the spatial heterogeneity implementation, the use of Perlin noise is appropriate but could be complemented by simpler configurations (e.g., two or four patches of different density), in that these simplified settings might permit analytical solutions.

Another point is about the experimental design vs. model assumptions, that there are several discrepancies between the model and experimental setup warrant discussion: 1. Model assumes heterogeneity in carrying capacity, while experiments directly set different cell densities; 2. Experimental wells lack direct contact, with manual mixing occurring hourly. How would model predictions change if simulated under exact experimental conditions?

Regarding the bioinformatics analysis, the genomic analysis is supportive but requires more careful interpretation, as there can be alternative explanations for observed correlations should be discussed For example, biofilm-forming microbes might have higher fitness, enabling them to maintain more MGEs, or MGEs themselves might carry genes promoting microbial clustering. These alternative mechanisms could be distinguished through additional analysis, and the relationship between spatial structure and MGE abundance might be bidirectional.

A minor comment is that the discussion section is overly lengthy and could be more focused.

Reviewer #3:

The manuscript MSB-2024-12666, entitled "Spatial entropy drives the maintenance and dissemination of mobile genetic elements" presented an interesting and important theoretical framework for studying the impact of spatial heterogeneity on the spread of mobile genetic elements (MGEs). By combining mathematical modeling, experimental validation, and analysis of gene flow in natural environments, the paper revealed how reducing spatial entropy can promote the persistence and dissemination of MGEs.

This research has significant theoretical implications and practical applications, particularly in managing the spread of antibiotic resistance genes (ARGs).

Major comments:

The paper directly refers to "patch" in both the Methods (line 386) and the first part of the Results section (line 78). For interdisciplinary researchers who may not be familiar with ecological modeling terminology, it is recommended to provide a brief explanation of what a "patch" represents in this context. This will help ensure the paper is accessible to a broader audience.

Additionally, other potentially confusing terms should be explained in a concise manner.

Although the authors described how spatial heterogeneity affects gene dissemination through variations in maximum carrying capacity, it is suggested to further elaborate on the biological background of across different patches. Specifically, why was a particular distribution chosen to represent? Is there empirical data supporting this specific heterogeneity setup?

The term "MGE-free bacteria" is used throughout the paper, but it would be helpful to clarify whether these bacteria are completely devoid of all types of mobile genetic elements, such as plasmids, transposons, or phages.

The manuscript would benefit from a clearer explanation regarding the purpose of calculating the maximum carrying capacity (N_m) for each patch and setting the average value to a constant (z). We understand that the likely intention behind keeping constant is to control for its influence and thus better isolate the effects of spatial entropy in the experiments. However, it would be helpful for the authors to explicitly state this purpose and discuss the rationale more clearly. Furthermore, the choice of specific parameters for determining N_m and z , as well as whether there is any empirical basis or objective reasoning behind this particular setting, is not sufficiently addressed.

The criteria used to define the boundaries or thresholds for low entropy and medium entropy conditions are not explicitly stated. It is crucial to clarify the specific values or ranges used to define these entropy categories, as this would help in understanding how these classifications were derived and ensure reproducibility of the study.

The manuscript proposed an insightful hypothesis that biofilm formation-related genes (BFGs) can serve as indicators of low entropy environments. I recommend that the authors provide the references or empirical data to strengthen this claim.

The definition of BFGs in the manuscript is based on matching the keyword "biofilm" in names. I recommend that the authors provide a comprehensive list of the BFGs used in their analysis. Additionally, it would be helpful to clarify if there are any genes known to participate in biofilm formation that might be overlooked by this keyword-based approach. Furthermore, please discuss if similar methodologies have been applied in previous studies on biofilm-related research and whether this approach is generally accepted in the field.

Point-by-point responses (in black) to reviewers' comments (in blue).

Reviewer #1:

The manuscript "Spatial entropy drives the maintenance and dissemination of mobile genetic elements" attempts to address a knowledge gap in the field concerning the influence of spatial landscape in maintenance and spread of mobile elements in environmental microbial communities. This study is important, timely and of significance as most of our current knowledge about MGE dynamics stems from studying bacteria and their MGEs in well-mixed populations, but these do not represent the spatial landscape in complex environmental communities.

The authors use a sound theoretical framework to represent microbial metacommunities in the environment and simulate gene flow dynamics. Based on the simulations where they keep cell densities constant while varying spatial entropy, the authors draw their main conclusion that lowering of spatial entropy favors maintenance and spread of mobile elements (MGE). The simulation methodology is rigorous and results based on them are convincing. The authors make a conceptual advance and also provide necessary context to explain their observations. They attribute their main conclusion to - resulting high local cell densities in low entropy environments, and establish relationship between spatial entropy and average rate of MGE transfer. They also attempt to validate this relationship based in theory using (1) an experimental set-up that measures MGE spread in *E. coli* metacommunities, and (2) through genome sequence analysis of bacteria from high and low spatial entropy environments. However, the manuscript falls short in providing convincing validation through their experimental and quantitative genomic analysis. Hence, the manuscript can be revised based on following suggestions.

We appreciate the reviewer's recognition of the importance, rigor and conceptual advance of this work. We are also grateful for the constructive comments and suggestions regarding further validations in experimental and genomic analyses. All the points raised have been fully addressed in the revised manuscript and are detailed below.

Major points -

1. Based on the description of *E. coli* metacommunities experiment the wells in each row represent a single meta-community. However, these wells are not connected i.e., a physical barrier separates members of the metacommunity. Given this, I would argue that the experiment does not represent the spatial landscape from the simulations described in Lines 82-85 - where the local patches are not separated by a barrier and allow for bacterial dispersal or immigration. Again, in the introduction Lines 51-53 the authors state that "the global maintenance of MGEs also depends on its efficiency to disseminate across different areas", however the experimental set-up does not support this condition. Hence, in my opinion the current experimental set-up with physical barriers is insufficient to validate the conditions used in the simulations.

We thank the reviewer for the insightful comments. To further validate the conditions used in the simulations, we conducted additional experiments in the updated manuscript. In these experiments, we connected adjacent wells in each row through periodic manual

pipetting. Every 15 minutes, 10 μ L of bacterial culture was transferred from each well to its neighboring wells and mixed thoroughly. This periodic mixing facilitated microbial dispersal across the patches. After one hour of incubation, the total number of transconjugants in each row was quantified using selective plating.

Our results indicated that microbial dispersal created by manual pipetting did not fundamentally alter our conclusion: reducing spatial entropy still promoted the effective plasmid transfer rate in the metacommunities.

The results of these additional experiments are discussed in lines 222–229 and presented in Appendix Fig. S9.

2. In support of their theory, the authors test the expectation that bacteria in low-entropy environments like biofilms should carry more MGEs compared to high entropy environments as low spatial entropy promotes MGE persistence and spread. To address this, they perform genome sequence analysis and stratify bacteria living in low entropy and high entropy environments based on presence or absence respectively of biofilm formation related genes (BFGs). The authors finds that bacteria with BFGs carry more plasmids compared to bacteria without BFGs. However, genome size is positively correlated with the number of MGEs(1,2). Hence, it is important to first determine that bacteria with BFGs do not have significantly larger genomes compared to bacteria without BFGs. This analysis is important to convincingly refute that the higher fraction of MGEs in the genomes with BFGs is not merely a consequence of their large genome sizes but is associated with low spatial entropy.

We are extremely grateful for the insightful suggestion. Distinguishing the role of entropy from the influence of genome size is crucial for justifying our conclusions. In response to the reviewer's comments, we first compared the genome sizes of bacteria with and without BFGs. We found that the genomes or chromosomes of bacteria with BFGs were significantly larger than those without BFGs (Appendix Fig. S12A and B).

To control for the effects of genome size, we divided our collection of prokaryotic genomes into multiple bins based on genome size. Within each bin, the size difference between the largest and smallest genomes was less than 2.4%, effectively minimizing the influence of genome size on plasmid carriage within each bin. We then analyzed whether the presence of BFGs led to a higher fraction of plasmid-carrying genomes in each bin. We found that in nearly all bins (13 out of 16), the fraction of plasmid-carrying genomes was higher among those with BFGs compared to those without BFGs (Appendix Fig. S12C-E).

These results suggest that the higher abundance of plasmids in genomes with BFGs is not merely a consequence of larger genome size but is associated with lower spatial entropy. More importantly, we found that the association between genome size and plasmid abundance exists only in genomes carrying BFGs. For genomes without BFGs, increasing genome size did not lead to a significant increase in plasmid number (Appendix Fig. S12C and D). This indicates that the relationship between genome size and plasmid number may be mediated by the capability of biofilm formation.

The results of this analysis are presented in Appendix Fig. S12 and discussed in line 258-275 in the updated manuscript.

Minor points -

1. Throughout the manuscript both in the simulations and experiments - the authors refer to the umbrella term of mobile elements or MGEs but the assumptions, environmental conditions and even the mechanisms specifically refer to MGEs that transfer via cell-to-cell contact, a particular MGE type i.e, plasmids. Hence, it might benefit to avoid generalisations and refer to mobile elements as plasmids throughout the manuscript.

We appreciate the reviewer's suggestion. Indeed, our work primarily focused on the process of plasmid transfer. We have replaced the general term 'mobile genetic elements' with the more specific term 'plasmids' throughout the updated manuscript.

2. In Figure 2a the scale requires a legend.

Thanks for pointing it out. We have added the legend of the scale in Fig. 2A.

3. Description of Figure 2a needs text describing within patch dynamics and description for the terms S_0 , S_1 , u_0 , u_1

We are grateful for the suggestion. In the updated legend of Fig. 2A, we have provided a more detailed description of the within-patch dynamics and have clarified the definitions of the four parameters (S_0 , S_1 , μ_0 and μ_1).

4. For Figure 4d and 5b, to improve readability the respective ends of x - axis could be marked high and low alongside their marked spatial entropy values. In addition, Line 450 from methods section could be added to the description text of Figure 4d for ease of understanding.

We appreciate the suggestions. In Fig. 4D and 5B, we have added a triangle bar along the x-axis to indicate high and low entropies. Additionally, we have incorporated the relevant text from line 450 of the Methods section into the description of Fig. 4D.

References:

1. I. L. G. Newton, S. R. Bordenstein, Correlations Between Bacterial Ecology and Mobile DNA. *Curr. Microbiol.* 62, 198-208 (2011).
2. S. Khedkar, G. Smyshlyaev, I. Letunic, O. M. Maistrenko, L. P. Coelho, A. Orakov, S. K. Forslund, F. Hildebrand, M. Luetge, T. S. B. Schmidt, O. Barabas, P. Bork, Landscape of mobile genetic elements and their antibiotic resistance cargo in prokaryotic genomes. *Nucleic Acids Res.* 50, 3155-3168 (2022).

We appreciate the reviewer for bringing two relevant papers to our attention. We have cited these works in the updated manuscript.

Reviewer #2:

This manuscript "Spatial entropy drives the maintenance and dissemination of mobile genetic elements" addresses fundamental questions regarding how spatial heterogeneity influences the maintenance and spread of mobile genetic elements (MGEs) in microbial communities. The authors present a novel theoretical framework that links spatial entropy to MGE persistence, supported by experimental validation and large-scale genomic analysis. This manuscript makes a significant contribution to understanding MGE maintenance and spread in spatially structured environments, and is particularly relevant given the growing concerns about antibiotic resistance and its spread through MGEs.

The study effectively bridges theoretical predictions with experimental validation and genomic evidence, and the introduction of spatial entropy as a unified measure for analyzing MGE persistence is innovative, providing implications for understanding and potentially controlling antibiotic resistance spread.

We are grateful for the reviewer's positive comments on the novelty, significance, and relevance of this work. We also appreciate the reviewer's suggestions regarding model assumptions, experimental design, and bioinformatic analysis, which have been instrumental in enhancing the rigor of our study. All the issues raised have been fully addressed in the updated manuscript and are detailed below.

However, there are some issues to be addressed, first is about the mathematical model assumptions. I think the relationship between density and transfer efficiency requires more thorough examination. The assumption that "gain of transfer efficiency by density increase exceeds the loss from local density reduction" may impose specific requirements on the "density-transfer efficiency" relationship. For example, the near-exponential increase in density-bacterial contact frequency (Figure 4c) is crucial for the results. What happens if the relationship saturates? Could density increases in certain regions fail to compensate for decreased transfer rates elsewhere? While the general conclusion that "spatial heterogeneity promotes coexistence" aligns with previous ecological theories, testing different functional forms would strengthen the mathematical foundation.

We are extremely grateful for the insightful comments. Indeed, the assumption that "gain of transfer efficiency by density increase exceeds the loss from local density reduction" stems from the *quadratic* relationship between effective transfer efficiency and local density. This relationship can be derived from the basic kinetics of plasmid transfer.

Let D , R and T represent the densities of plasmid donors, recipients and transconjugants, respectively. The reaction order (denoted as σ) of plasmid transfer describes the relationship between the rate of transconjugant production and the concentrations of donors and recipients:

$$\frac{d[T]}{dt} = \eta[D]^\sigma[R]^\sigma.$$

Let C be the overall cell density in the local population, and ϑ be the ratio between $[D]$

and [R]. The rate of plasmid transfer can then be expressed as:

$$\frac{d[T]}{dt} = \eta \left(\frac{\vartheta}{1 + \vartheta} \right)^\sigma \left(\frac{1}{1 + \vartheta} \right)^\sigma [C]^{2\sigma}.$$

Therefore, the relationship between transfer efficiency and cell density is determined by the value of σ :

$$\frac{d[T]}{dt} \propto [C]^{2\sigma}.$$

The value of σ dictates whether the gain of transfer efficiency from increased density outweighs the loss from local density reduction. Specifically, when $\sigma > 0.5$, density increases in certain regions can compensate for decreased transfer rates elsewhere. When $\sigma = 0$, corresponding to zero-order kinetics, the relationship between transfer efficiency and cell density saturates.

Classic plasmid ecological theories commonly employ a mass-action model, where single cells randomly collide and exchange plasmids upon collision, leading to first-order kinetics of plasmid transfer ($\sigma = 1$). In our simulations, we also assumed $\sigma = 1$, consistent with previous studies. Therefore, the plasmid transfer efficiency becomes quadratically related to cell density:

$$\frac{d[T]}{dt} \propto [C]^2.$$

In this case, the gain in transfer efficiency from increased density exceeds the loss from local density reduction.

In light of the reviewer's comments, we have conducted additional simulations to evaluate how our general conclusions are influenced by varying the relationship between transfer efficiency and cell density. Specifically, we repeated the patch dynamics simulations under three conditions for σ :

- (1) **When $\sigma < 0.5$:** The gain in contact frequency from increased density does not compensate for the loss from local density reduction. In this case, reducing spatial entropy slows down the accumulation rate of a transferable plasmid (Appendix Fig. S8A).
- (2) **When $\sigma = 0.5$:** The gain in contact frequency from increased density equals the loss from local density reduction. In this case, reducing spatial entropy does not influence the accumulation rate of a transferable plasmid (Appendix Fig. S8B).
- (3) **When $\sigma > 0.5$:** The gain in contact frequency from increased density exceeds the loss from local density reduction. In this case, reducing spatial entropy promotes the accumulation rate of a transferable plasmid (Appendix Fig. S8C and D).

We also considered a scenario where cell contact frequency is a Hill function of bacterial density (Appendix Fig. S8E). In this case, the reaction order σ of plasmid transfer transitions from 1 to 0 as density increases. Our simulation results suggested that

reducing spatial entropy slows down the accumulation rate of a transferable plasmid in this scenario.

These results align with the reviewer's comments and highlight that the functional form of the density-bacterial contact frequency relationship indeed affects the interplay between spatial entropy and plasmid maintenance. While various mathematical forms can be explored, the general conclusion that 'spatial heterogeneity promotes plasmid maintenance', validated by experiments and bioinformatic analysis, suggested that $\sigma > 0.5$ is more realistic. This assumption is also consistent with classic plasmid ecological theories which commonly use $\sigma = 1$ for plasmid transfer kinetics.

The results of the above analysis were presented in Appendix Fig. S8 and discussed in Appendix Text S3.

Also, about the spatial heterogeneity implementation, the use of Perlin noise is appropriate but could be complemented by simpler configurations (e.g., two or four patches of different density), in that these simplified settings might permit analytical solutions.

We appreciate the suggestion. In the updated manuscript, we have conducted additional simulations focusing on a simplified system of two patches. By varying the cell density ratio between these two patches, we were able to control the spatial entropy of the system. Our simulations consistently showed that reducing spatial entropy enhances the overall abundance of the plasmid, regardless of the plasmid transfer rate or the mean cell density. These results are presented in Appendix Fig. S5 and discussed in lines 161-163 of the updated manuscript.

Even in this simplified system with only two patches, the mathematical model involves four ordinary differential equations (ODEs), making an analytical solution impractical. To address this, we adopted the concept of plasmid persistence potential and its quantitative relationship with plasmid abundance. This approach allowed us to derive the analytical solution for plasmid abundances within the two patches by solving a complex cubic equation. The detailed analytical derivation is provided in Appendix Text S1.

Although we were able to formulate the analytical solution, its complexity limits its ability to provide clear and intuitive insights. In such cases, mathematical simulations are more effective in revealing the interplay between spatial entropy and plasmid abundance.

Another point is about the experimental design vs. model assumptions, that there are several discrepancies between the model and experimental setup warrant discussion: 1. Model assumes heterogeneity in carrying capacity, while experiments directly set different cell densities; 2. Experimental wells lack direct contact, with manual mixing occurring hourly. How would model predictions change if simulated under exact experimental conditions?

We thank the reviewer for the insightful comments. The reviewer raised two major discrepancies between the model and experimental conditions.

(1) **Control of cell density**

In our previous models, we controlled the cell density of each patch by adjusting the maximum carrying capacity. This approach was based on the observation that, at low dilution rates, the steady-state density of a population closely approximates its maximum carrying capacity. To evaluate whether this discrepancy was critical, we developed a simplified model where the population density remained constant during plasmid transfer. In this model, the spatial entropy of the metacommunity can be directly controlled by setting different cell densities for each patch.

(2) **Discrete vs. continuous dispersal**

Our previous model assumed continuous dispersal among patches. To test whether this assumption affected our predictions, we considered an array of patches undergoing periodic mixing. In this setup, local populations in each patch grow separately without cross-patch dispersal until the next mixing event. This approach more closely mimics the discrete nature of dispersal in experiments.

By combining density-constant population dynamics and periodic mixing, we conducted additional simulations that better reflected experimental conditions. We controlled the spatial entropy of the system by directly manipulating the density distributions across patches. After a given number of mixing steps, we calculated the entropy value and plasmid abundance. Our results consistently showed that reducing spatial entropy promotes plasmid persistence and abundance, even under these modified conditions. These findings underscore the robustness of our predictions and highlight the importance of spatial entropy across diverse scenarios.

The details of these supplementary simulations are described in Appendix Text S2. The results are presented in Appendix Fig. S6 and discussed in lines 163-168 of the updated manuscript.

Regarding the bioinformatics analysis, the genomic analysis is supportive but requires more careful interpretation, as there can be alternative explanations for observed correlations should be discussed. For example, biofilm-forming microbes might have higher fitness, enabling them to maintain more MGEs, or MGEs themselves might carry genes promoting microbial clustering. These alternative mechanisms could be distinguished through additional analysis, and the relationship between spatial structure and MGE abundance might be bidirectional.

We appreciate the reviewer for raising two alternative explanations that require more careful analysis and discussion.

(1) **Biofilm-forming microbes might have higher fitness, enabling them to maintain more MGEs.**

Indeed, the biofilms may provide several advantages that enhance microbial fitness, such as physical protection against environmental stresses and optimized resource acquisition (Flemming *et al.*, 2016). However, the relative fitness of biofilm-dwelling microbes compared to their free-living counterparts can vary significantly depending on the microbial species and environmental conditions (Flemming *et al.*, 2016). For example, in

nutrient-scarce environments, biofilm microbes may have less access to resources than free-living cells that can move more freely. Similarly, in low-oxygen environments, aerobic microbes in biofilms may be disadvantaged compared to free-living cells that can access oxygen more readily. Biofilms can also accumulate toxins or waste products that are harmful to the microbes within them (Bonnineau *et al*, 2021). Given these complex contexts, we recognize that genomic sequences alone may be insufficient to reliably estimate microbial fitness in or out of biofilms. Disentangling the effects of microbial fitness would likely require more in-depth quantification of microbial growth dynamics in real environments.

(2) MGEs themselves might carry genes promoting microbial clustering

To evaluate the contributions of plasmids to biofilm formation, we calculated the number of biofilm-forming genes (BFGs) carried by each plasmid. Among the 34,688 genomes we analyzed, 11,939 (34.4%) carried at least one BFG. However, only a small fraction (1.8%) of these genomes had BFGs located on plasmids. In total, we identified 71,522 BFGs across all genomes, with plasmids contributing only 278 genes (~0.39%). Thus, most genes promoting microbial clustering are chromosomally encoded, and the contribution of plasmids to biofilm formation appears to be relatively minor. To further assess the impact of plasmid-borne BFGs on our conclusions, we repeated our analysis after excluding genomes containing BFGs on plasmids. The results still indicated that biofilm formation promotes the maintenance of plasmids in prokaryotic genomes, even when plasmids do not contribute to biofilm formation. These findings suggest that the observed correlation between biofilm formation capability and plasmid abundance is likely mediated by entropy effects rather than direct contributions from MGEs.

The results of this supplementary analysis are presented in Appendix Figs. S18 and S19, and we have also discussed these alternative explanations in the updated manuscript (line 344-371).

A minor comment is that the discussion section is overly lengthy and could be more focused.

We appreciate the comment. The discussion section of the updated manuscript has been substantially streamlined for clarity and conciseness.

Reviewer #3:

1. The manuscript MSB-2024-12666, entitled "Spatial entropy drives the maintenance and dissemination of mobile genetic elements" presented an interesting and important theoretical framework for studying the impact of spatial heterogeneity on the spread of mobile genetic elements (MGEs). By combining mathematical modeling, experimental validation, and analysis of gene flow in natural environments, the paper revealed how reducing spatial entropy can promote the persistence and dissemination of MGEs. This research has significant theoretical implications and practical applications, particularly in managing the spread of antibiotic resistance genes (ARGs).

We are grateful to the reviewer for recognizing the significance and interest of this work. We also appreciate the constructive comments regarding the clarification of key concepts and assumptions. All the issues raised have been thoroughly addressed in the updated manuscript, as detailed below.

2. Major comments:

The paper directly refers to "patch" in both the Methods (line 386) and the first part of the Results section (line 78). For interdisciplinary researchers who may not be familiar with ecological modeling terminology, it is recommended to provide a brief explanation of what a "patch" represents in this context. This will help ensure the paper is accessible to a broader audience. Additionally, other potentially confusing terms should be explained in a concise manner.

We appreciate the suggestion. In the updated manuscript, we have provided detailed explanations of these concepts (line 87-96). Specifically, we have clarified that in microbial ecology, a metacommunity is defined as a collection of local communities distributed across different habitat patches. Each patch represents a discrete, relatively small area within a larger landscape that supports a local population. Patches serve as the fundamental spatial units where ecological processes unfold, and they are often characterized by the unique environmental conditions or resources that can influence the composition and structure of the community residing within them. Patches are interconnected through the dispersal routes of species, facilitating the exchange of organisms and genetic materials across the metacommunity.

3. Although the authors described how spatial heterogeneity affects gene dissemination through variations in maximum carrying capacity, it is suggested to further elaborate on the biological background of across different patches. Specifically, why was a particular distribution chosen to represent? Is there empirical data supporting this specific heterogeneity setup?

We are grateful for the comments. In our study, we employed Perlin noise-like distributions to model the diverse spatial landscapes of metacommunities. Although empirical data on heterogeneity patterns in real environments are limited, we believe that the relationship between spatial entropy and plasmid maintenance is not dependent on any specific heterogeneity setup.

Considering the reviewer's comment, we have expanded our analysis in the updated

manuscript. Specifically, we tested additional heterogeneity setups generated using 2D Gaussian and 2D uniform distribution algorithms. We then simulated plasmid transfer dynamics within these metacommunities. Our results consistently showed that reducing spatial entropy enhances plasmid maintenance, regardless of the heterogeneity setup. This confirms that the effect of spatial entropy is independent of the specific spatial structure of the metacommunities.

The additional simulations are described in Appendix Text S4. The results are presented in Appendix Fig. S7 and discussed in lines 171–178.

4. The term "MGE-free bacteria" is used throughout the paper, but it would be helpful to clarify whether these bacteria are completely devoid of all types of mobile genetic elements, such as plasmids, transposons, or phages.

We appreciate the comments. In this study, our primary focus was on the process of plasmid transfer. Accordingly, when we referred to a bacterial cell as 'MGE-free' we specifically meant that the cell did not carry the transferable plasmid in question. However, it is possible that the cell could still harbor other types of mobile genetic elements (MGEs), such as transposons or phages. To avoid any potential confusions, we have revised the terminology in the updated manuscript, replacing 'MGE-free bacteria' with 'plasmid-free bacteria'.

5. The manuscript would benefit from a clearer explanation regarding the purpose of calculating the maximum carrying capacity (N_m) for each patch and setting the average value to a constant (z). We understand that the likely intention behind keeping constant is to control for its influence and thus better isolate the effects of spatial entropy in the experiments. However, it would be helpful for the authors to explicitly state this purpose and discuss the rationale more clearly. Furthermore, the choice of specific parameters for determining N_m and z , as well as whether there is any empirical basis or objective reasoning behind this particular setting, is not sufficiently addressed.

We are grateful for the comments. In the updated manuscript, we have explicitly explained the rationale behind calculating the maximum carrying capacity for each patch and setting the average value to a constant (line 141-149, 152-154, 337-343).

In a metacommunity of $m \times m$ patches, we calculated the maximum carrying capacity N_m for the patch $[i, j]$ as $z \frac{|a_{ij}|^n}{\sum_{i,j} |a_{ij}|^n} \cdot m^2$. This ensures that the mean value of N_m across all patches equals z . The mean cell density in the metacommunity can thus be controlled by adjusting z . a_{ij} is a random number generated using Perlin noise, $|a_{ij}|$ denotes its absolute value and n is an integer. The value of n controls the controls the spatial fluctuation of bacterial density, with larger n values leading to lower spatial entropy. When $n = 0$, all N_m values become identical, resulting in a homogeneous distribution of microbes across the metacommunity.

In the initial manuscript, the parameters for N_m and z were generated using Perlin noise algorithm to create specific spatial landscapes within the metacommunities. While empirical data on heterogeneity patterns in real environments are limited, we have

demonstrated that our conclusions are robust and do not rely on specific parameter settings. Specifically, we tested different heterogeneity setups generated by either 2D Gaussian or 2D uniform distribution algorithms. Our simulations indicated that altering these heterogeneity settings did not affect our conclusions. For more details, please refer to our response to point 3.

6. The criteria used to define the boundaries or thresholds for low entropy and medium entropy conditions are not explicitly stated. It is crucial to clarify the specific values or ranges used to define these entropy categories, as this would help in understanding how these classifications were derived and ensure reproducibility of the study.

We appreciate the comments. In Fig. 4A, we depicted three bacterial populations arranged from left to right in decreasing order of spatial entropy. It should be noted that we did not apply any specific criteria or thresholds to categorize them as “high entropy,” “medium entropy,” and “low entropy.” These labels were merely used to illustrate the relative ranking of the populations. To avoid any potential confusion, we have removed these labels in the updated manuscript and added a triangle bar to clearly indicate the decreasing spatial entropy from left to right. In Fig. 4B, we displayed the steady-state plasmid abundance in the metacommunity as a function of plasmid transfer rate η , under different spatial entropies. The curves under three entropy conditions were shown as examples. In the updated manuscript, we have also explicitly stated the entropy values corresponding to the examples of ‘low entropy’ and ‘medium entropy’ conditions (line 616-617).

7. The manuscript proposed an insightful hypothesis that biofilm formation-related genes (BFGs) can serve as indicators of low entropy environments. I recommend that the authors provide the references or empirical data to strengthen this claim.

We sincerely appreciate the reviewer’s positive feedback and valuable suggestions. This claim is supported by the following studies:

- Kamali E *et al.* examined the biofilm phenotype and the presence of biofilm-related genes in 80 clinical isolates of *Pseudomonas aeruginosa* (Kamali *et al.*, 2020). Their study revealed a highly significant correlation between biofilm-forming capacity and the presence of biofilm-related genes (p-value < 0.0001).
- Bostanghadiri N *et al.* investigated biofilm production and the presence of biofilm genes in clinical isolates of *Stenotrophomonas maltophilia* (Bostanghadiri *et al.*, 2021). Biofilm formation-related genes were detected in all biofilm-producing isolates, and their presence significantly enhanced biofilm production.
- A similar study by Zhuo C *et al.* also demonstrated a strong association between biofilm formation and the presence of these genes (Zhuo *et al.*, 2014).

Collectively, these studies suggest that the presence of biofilm formation-related genes can serve as a reliable indicator of a bacterium’s capacity to form biofilms. Compared to free-swimming planktonic cells, bacterial biofilms exhibit greater spatial heterogeneity, or lower spatial entropy. This aspect has been extensively reviewed in several papers

(Stewart & Franklin, 2008; Wimpenny *et al*, 2000).

In light of the reviewer's suggestion, we have incorporated the above references in the revised manuscript.

8. The definition of BFGs in the manuscript is based on matching the keyword "biofilm" in names. I recommend that the authors provide a comprehensive list of the BFGs used in their analysis. Additionally, it would be helpful to clarify if there are any genes known to participate in biofilm formation that might be overlooked by this keyword-based approach. Furthermore, please discuss if similar methodologies have been applied in previous studies on biofilm-related research and whether this approach is generally accepted in the field.

We are grateful for the constructive feedback. In the revised manuscript, we have included a comprehensive list of the BFGs utilized in our analysis (Appendix Table S2). To minimize the possibility that our approach might overlook any genes known to be involved in biofilm formation, we conducted an extensive search using the NCBI Gene database, which provides detailed gene sequences, functional annotations, and related pathways. By filtering with the term 'biofilm', we identified a total of 124 genes associated with biofilm formation (Appendix Table S3, line 311-317). Notably, 120 of these genes (96.8%) were already included in our initial list of BFGs, indicating a high degree of coverage in our original analysis.

To further expand our search scope, we employed additional filters, including 'adhesion', 'quorum sensing', 'extracellular matrix', 'exopolysaccharide biosynthesis', and 'exopolysaccharide production' (Appendix Tables S4–S7, line 317-325). These terms are closely related to biofilm formation. Using these filters, we generated lists containing 19, 39, 10, 27, and 9 genes, respectively. By integrating these newly identified genes with our previously recognized ones, we created an extended list of biofilm formation-related genes, comprising 240 non-redundant genes. To the best of our knowledge, this extended list encompasses most genes involved in biofilm formation, including the *eps*, *psl*, and *sia* gene families (Chen *et al*, 2020; Marvasi *et al*, 2010; Overhage *et al*, 2005).

We then repeated our analysis using this expanded list of BFGs. The results indicated that even with the inclusion of these additional genes, the presence of BFGs continued to promote the carriage and abundance of plasmids in prokaryotic genomes (Appendix Fig. S17, line 325-328). This finding underscores the robustness of our conclusions regarding the coverage of biofilm-related genes.

We acknowledge that a keyword-based approach has been utilized in previous biofilm-related studies. For example, Magalhães *et al*. conducted searches across various databases, including PubMed, using biofilm-related keywords to gather information on proteins and enzymes involved in biofilm formation (Magalhães *et al*, 2020). Similarly, Zhang *et al*. employed text-mining analysis with the keyword 'biofilm' to identify biofilm-associated proteins (Zhang *et al*, 2024). These studies have been cited and discussed in the revised manuscript (line 306-311).

References

- Bonnineau C, Artigas J, Chaumet B, Dabrin A, Faburé J, Ferrari BJ, Lebrun JD, Margoum C, Mazzella N, Miege C (2021) Role of biofilms in contaminant bioaccumulation and trophic transfer in aquatic ecosystems: current state of knowledge and future challenges. *Reviews of Environmental Contamination and Toxicology* 253: 115-153
- Bostanghadiri N, Ardebili A, Ghalavand Z, Teymouri S, Mirzarazi M, Goudarzi M, Ghasemi E, Hashemi A (2021) Antibiotic resistance, biofilm formation, and biofilm-associated genes among *Stenotrophomonas maltophilia* clinical isolates. *BMC Research Notes* 14: 1-6
- Chen G, Gan J, Yang C, Zuo Y, Peng J, Li M, Huo W, Xie Y, Zhang Y, Wang T (2020) The SiaA/B/C/D signaling network regulates biofilm formation in *Pseudomonas aeruginosa*. *The EMBO Journal* 39: e103412
- Flemming H-C, Wingender J, Szewzyk U, Steinberg P, Rice SA, Kjelleberg S (2016) Biofilms: an emergent form of bacterial life. *Nature Reviews Microbiology* 14: 563-575
- Kamali E, Jamali A, Ardebili A, Ezadi F, Mohebbi A (2020) Evaluation of antimicrobial resistance, biofilm forming potential, and the presence of biofilm-related genes among clinical isolates of *Pseudomonas aeruginosa*. *BMC Research Notes* 13: 1-6
- Magalhães RP, Vieira TF, Fernandes HS, Melo A, Simões M, Sousa SF (2020) The biofilms structural database. *Trends in Biotechnology* 38: 937-940
- Marvasi M, Visscher PT, Casillas Martinez L (2010) Exopolymeric substances (EPS) from *Bacillus subtilis*: polymers and genes encoding their synthesis. *FEMS Microbiology Letters* 313: 1-9
- Overhage Jr, Schemionek M, Webb JS, Rehm BH (2005) Expression of the *psl* operon in *Pseudomonas aeruginosa* PAO1 biofilms: PslA performs an essential function in biofilm formation. *Applied and Environmental Microbiology* 71: 4407-4413
- Stewart PS, Franklin MJ (2008) Physiological heterogeneity in biofilms. *Nature Reviews Microbiology* 6: 199-210
- Wimpenny J, Manz W, Szewzyk U (2000) Heterogeneity in biofilms. *FEMS Microbiology Reviews* 24: 661-671
- Zhang Z, Pan Y, Hussain W, Chen G, Li E (2024) BBSdb, an open resource for bacterial biofilm-associated proteins. *Frontiers in Cellular and Infection Microbiology* 14: 1428784
- Zhuo C, Zhao Q-y, Xiao S-n (2014) The impact of *spgM*, *rpfF*, *rmlA* gene distribution on biofilm formation in *Stenotrophomonas maltophilia*. *PLoS One* 9: e108409

26th Mar 2025

Manuscript Number: MSB-2024-12666R

Title: Spatial entropy drives the maintenance and dissemination of transferable plasmids

Dear Prof Wang,

Thank you for the submission of your revised manuscript to Molecular Systems Biology. We have now received the enclosed reports from the referees that were asked to re-assess it. As you will see the reviewers are now globally supportive and I am pleased to inform you that we will be able to accept your manuscript pending the following final amendments:

1) In the main manuscript file, please include keywords to max. 5.

2) Please format the Data availability section according to the example below:

"The datasets and computer code produced in this study are available in the following databases:

- Chip-Seq data: Gene Expression Omnibus GSE46748 (<https://www.ncbi.nlm.nih.gov/geo/query/acc.cgi?acc=GSE46748>)

- Modeling computer scripts: GitHub (<https://github.com/SysBioChalmers/GECKO/releases/tag/v1.0>)

- [data type]: [full name of the resource] [accession number/identifier] ([doi or URL or identifiers.org/DATABASE:ACCESSION])"

3) Code: Please update the README file on Github with practical use instructions for potential future users of your code. In addition, we are not able to open the PDFs loaded in Github ('Error rendering embedded code. Invalid PDF')

4) Author contributions: Please remove it from the manuscript and specify author contributions in our submission system.

CRedit has replaced the traditional author contributions section because it offers a systematic machine-readable author contributions format that allows for more effective research assessment. You are encouraged to use the free text boxes beneath each contributing author's name to add specific details on the author's contribution. More information is available in our guide to authors:

<https://www.embopress.org/page/journal/17574684/authorguide#authorshipguidelines>

5) Our journal encourages inclusion of *data citations in the reference list* to directly cite datasets that were re-used and obtained from public databases. Data citations in the article text are distinct from normal bibliographical citations and should directly link to the database records from which the data can be accessed. In the main text, data citations are formatted as follows: "Data ref: Smith et al, 2001" or "Data ref: NCBI Sequence Read Archive PRJNA342805, 2017". In the Reference list, data citations must be labeled with "[DATASET]". A data reference must provide the database name, accession number/identifiers and a resolvable link to the landing page from which the data can be accessed at the end of the reference. Further instructions are available at .

6) Please ensure that a statement on whether or not blinding was done is included in the Methods even if no blinding was done. Please also be sure to update the Author Checklist with this information and where it can be found in the manuscript.

7) Please remove the Reagents and Tools Table from the Methods section of the manuscript as you have already uploaded it as a separate file.

8) Please place individual sections of the manuscript in the following order: Title page - Abstract & Keywords - Introduction - Results - Discussion - Methods - Data Availability - Acknowledgements - Disclosure and Competing Interests Statement - References - Figure Legends - Expanded View Figure Legends.

9) For the figures and figure legends, please take care of the following:

- Please indicate what */ **/ ***/ **** represents; if this represents p value(s), please indicate the statistical test used and where appropriate, and the exact p value in the legend(s) or directly in the figures of figure(s) 6C, D, E, F, G, H, J, K, L, M.

- Please note that the box plots need to be defined in terms of minima, maxima, centre, bounds of box and whiskers, and percentile in the legends of figures 6C, D, E, F, G, H, J, K, L, M

- Please note that information related to n is missing in the legends of figures 6C, D, E, F, G, H, J, K, L, M

- In a routine figure check, we note that parts of Figure 3A overlap with parts of Appendix Figure S2A as it appears that the same examples of landscapes across different mean densities, spatial entropies, and spatial periods was examined, which is perfectly acceptable. However, for transparency for the readers, we would ask you to indicate that the same information was used to create both figures in the respective figure legends.

10) Appendix file: Please add page numbers to the Table of Contents.

11) Synopsis image: Please provide the synopsis image as a high-resolution jpeg or PNG file (not PDF) with dimensions 550 pixels wide x (300-600) pixels high.

12) As part of the EMBO Publications transparent editorial process initiative (see our policy here:

https://www.embopress.org/transparent-process#Review_Process), Molecular Systems Biology will publish online a Peer Review File (PRF) to accompany accepted manuscripts. This file will be published in conjunction with your paper and will include the anonymous referee reports, your point-by-point response and all pertinent correspondence relating to the manuscript. Let us know whether you agree with the publication of the PRF and as here, if you want to remove or not any figures from it prior to publication. Please note that the Authors checklist will be published at the end of the PRF.

13) After your paper is published, we will promote it on social media. If you have any handles or hashtags for Bluesky you would like included, please let us know.

14) Please provide a point-by-point letter INCLUDING my comments and your detailed responses (as Word file).

I look forward to reading a new revised version of your manuscript as soon as possible.

Yours sincerely,

Poonam Bheda, PhD
Scientific Editor
Molecular Systems Biology

Reviewer #1:

The authors have addressed major concerns regarding the experimental and quantitative genomic validations in support of the results from simulations. The additional experiments, analysis and their appropriate inclusion in the manuscript text and figures provide necessary evidence supporting the claims made in the manuscript. I have no further suggestions and support the publication of this manuscript.

(comments on author responses to Reviewer 3):

I have now gone through the responses to concerns raised by Reviewer 3. In my opinion, for majority of the points the authors have sufficiently addressed the concerns. However, points 3 and 5 from Reviewer 3 are out my field of work/expertise and hence I cannot judge whether if these are sufficiently addressed by the authors.

Given this, for all points from reviewer 3 (except for 3 and 5) my response would be - the authors have sufficiently addressed the raised concerns.

Reviewer #2:

This round of revision has satisfactorily addressed my previous concerns. The new experiments, modeling approaches, and data analyses adequately support the authors' conclusions and have significantly improved the manuscript's logical flow and clarity. The paper now presents a coherent and compelling scientific narrative. I recommend publication without further revision

(thoughts on the author's response to #3 reviewer's comments)

From what I can see, the #3 reviewer's comments are mostly about writings: they requested clarification on technical terms like "patch" for interdisciplinary readers, suggested clearer explanation of parameters like maximum carrying capacity, recommended better definition of entropy thresholds, and questioned the biological rationale behind the chosen heterogeneity distribution. I think these concerns are mostly about writing instead of the scientific validity and novelty of this work itself.

In authors' responses, the authors thoroughly addressed these concerns by adding explanations of ecological concepts, testing additional heterogeneity setups to prove their findings were robust across different spatial structures, replacing "MGE-free bacteria" with "plasmid-free bacteria" for precision, clarifying their rationale for parameter choices, removing potentially confusing entropy labels, and bolstering their biofilm formation hypothesis with substantial references. They also expanded their analysis of biofilm-formation related genes beyond keyword matching, compiling a comprehensive gene list and confirming that their findings remained consistent even with this expanded dataset.

Therefore, I think the author has sufficiently addressed all concerns raised by the third reviewer.

Point-by-point responses (in black) to editorial requests (in blue).

1) In the main manuscript file, please include keywords to max. 5.

In the updated manuscript file, we have included the following keywords: horizontal gene transfer, plasmid, spatial entropy, antibiotic resistance, biofilm.

2) Please format the Data availability section according to the example below:

"The datasets and computer code produced in this study are available in the following databases:

- Chip-Seq data: Gene Expression Omnibus GSE46748

(<https://www.ncbi.nlm.nih.gov/geo/query/acc.cgi?acc=GSE46748>)

- Modeling computer scripts: GitHub

(<https://github.com/SysBioChalmers/GECKO/releases/tag/v1.0>)

- [data type]: [full name of the resource] [accession number/identifier] ([doi or URL or identifiers.org/DATABASE:ACCESSION])"

We have updated the Data availability section to align with the requested format.

3) Code: Please update the README file on Github with practical use instructions for potential future users of your code. In addition, we are not able to open the PDFs loaded in Github ('Error rendering embedded code. Invalid PDF')

We have updated the README file on Github with practical use instructions for potential future users. To verify document accessibility, we have checked whether the PDFs could be loaded on at least three different computers. On all the computers the PDFs were successfully loaded in Github. Based on these observations, we hypothesize that the loading failures might be attributable to network connectivity issues or browser-specific compatibility challenges.

4) Author contributions: Please remove it from the manuscript and specify author contributions in our submission system. CRediT has replaced the traditional author contributions section because it offers a systematic machine-readable author contributions format that allows for more effective research assessment. You are encouraged to use the free text boxes beneath each contributing author's name to add specific details on the author's contribution. More information is available in our guide to authors:

<https://www.embopress.org/page/journal/17574684/authorguide#authorshipguidelines>

We have removed Authors contributions from the manuscript and specified it in the submission system.

5) Our journal encourages inclusion of *data citations in the reference list* to directly cite datasets that were re-used and obtained from public databases. Data citations in the article text are distinct from normal bibliographical citations and should directly link to the database records from which the data can be accessed. In the main text, data citations are formatted as follows: "Data ref: Smith et al, 2001" or "Data ref: NCBI Sequence Read Archive PRJNA342805, 2017". In the Reference list, data citations must be labeled with "[DATASET]". A data reference must provide the database name, accession

number/identifiers and a resolvable link to the landing page from which the data can be accessed at the end of the reference. Further instructions are available at <https://www.embopress.org/page/journal/17574684/authorguide#referencesformat>.

We have included the data citations in the reference list to directly cite the RefSeq database.

6) Please ensure that a statement on whether or not blinding was done is included in the Methods even if no blinding was done. Please also be sure to update the Author Checklist with this information and where it can be found in the manuscript.

We have included the following statement in Methods: ‘We did not employ randomization procedures for sample allocation, nor did we implement blinding during experiments or data analysis’. We have also updated the Author Checklist with this information.

7) Please remove the Reagents and Tools Table from the Methods section of the manuscript as you have already uploaded it as a separate file.

We have removed the Reagents and Tools Table from the Methods section.

8) Please place individual sections of the manuscript in the following order: Title page - Abstract & Keywords - Introduction - Results - Discussion - Methods - Data Availability - Acknowledgements - Disclosure and Competing Interests Statement - References - Figure Legends - Expanded View Figure Legends.

We have placed individual sections of the manuscript in the requested order.

9) For the figures and figure legends, please take care of the following:

- Please indicate what */ **/ ***/ **** represents; if this represents p value(s), please indicate the statistical test used and where appropriate, and the exact p value in the legend(s) or directly in the figures of figure(s) 6C, D, E, F, G, H, J, K, L, M.

In the updated figure legends, we have explicitly stated that “the triple asterisk notation (***) denotes statistical significance at the $p < 0.001$ level, as determined by two-sided Student's t-tests”. Where applicable, we have included exact p-values in the figure legends. For certain analyses, the obtained p-values were below the detection threshold of our statistical software ($p = 0.0$ due to automatic rounding). In accordance with standard scientific practice, these values have been reported as $p < 0.001$.

- Please note that the box plots need to be defined in terms of minima, maxima, centre, bounds of box and whiskers, and percentile in the legends of figures 6C, D, E, F, G, H, J, K, L, M

We have defined the box plots in terms of minima, maxima, centre, bounds of box and whiskers, and percentile in the updated legends.

- Please note that information related to n is missing in the legends of figures 6C, D, E, F, G, H, J, K, L, M

We have added the information related to n in the updated legends.

- In a routine figure check, we note that parts of Figure 3A overlap with parts of Appendix Figure S2A as it appears that the same examples of landscapes across different mean densities, spatial entropies, and spatial periods was examined, which is perfectly acceptable. However, for transparency for the readers, we would ask you to indicate that the same information was used to create both figures in the respective figure legends.

Yes, the middle columns in both Fig. 3A and Fig. S2A utilize identical landscape examples. In Fig. 3A, the left-to-right progression of columns represents landscapes with varying spatial periods, whereas in Fig. S2A, this progression corresponds to landscapes with different mean densities. We have explicitly noted in both figure legends that the middle columns share the same underlying data.

10) Appendix file: Please add page numbers to the Table of Contents.

We have added the page numbers to the Table of Contents.

11) Synopsis image: Please provide the synopsis image as a high-resolution jpeg or PNG file (not PDF) with dimensions 550 pixels wide x (300-600) pixels high.

We have provided the synopsis image as a high-resolution PNG file.

12) As part of the EMBO Publications transparent editorial process initiative (see our policy here: https://www.embopress.org/transparent-process#Review_Process), Molecular Systems Biology will publish online a Peer Review File (PRF) to accompany accepted manuscripts. This file will be published in conjunction with your paper and will include the anonymous referee reports, your point-by-point response and all pertinent correspondence relating to the manuscript. Let us know whether you agree with the publication of the PRF and as here, if you want to remove or not any figures from it prior to publication. Please note that the Authors checklist will be published at the end of the PRF.

We agree with the publication of the PRF. No figures need to be removed from PRF prior to publication.

13) After your paper is published, we will promote it on social media. If you have any handles or hashtags for Bluesky you would like included, please let us know.

Thanks. We don't have any handles or hashtags for Bluesky to be included.

14) Please provide a point-by-point letter INCLUDING my comments and your detailed responses (as Word file).

We have provided a point-by-point letter including editorial comments and our detailed responses as a Word file.

11th Apr 2025

Manuscript number: MSB-2024-12666RR

Title: Spatial entropy drives the maintenance and dissemination of transferable plasmids

Dear Prof Wang,

Thank you again for sending us your revised manuscript. We are now satisfied with the modifications made and I am pleased to inform you that your paper has been accepted for publication.

Yours sincerely,

Poonam Bheda, PhD
Scientific Editor
Molecular Systems Biology
